# The Ice Cloud Imager: retrieval of frozen water mass profiles

Eleanor May<sup>1</sup> and Patrick Eriksson<sup>1</sup>

<sup>1</sup>Department of Space, Earth and Environment, Chalmers University of Technology, Gothenburg, Sweden

**Correspondence:** Eleanor May (eleanor.may@chalmers.se)

## Abstract.

The Ice Cloud Imager (ICI) will be hosted on the second generation of the EUMETSAT Polar System (EPS-SG). By measuring at microwave and sub-millimetre wavelengths, ICI will provide unparalleled global observations of ice clouds. EUMETSAT's official ICI level-2 product will offer retrievals of ice mass column properties. This study explores whether the capabilities of ICI can be extended to retrieve vertical profiles of ice mass.

Using a retrieval database of ICI simulations, we trained a quantile regression neural network (QRNN) to retrieve ice water content (IWC) and profiles of the mean mass diameter of ice hydrometeors. Our retrieval setup is fast and simpler to implement than previous ICI profile retrieval approaches, and the study is more comprehensive in scope than earlier efforts. Comparisons between our retrieved and database profiles demonstrate that ICI observations are sensitive to IWC within the range of  $10^{-2}$  and  $1~{\rm g~m^{-3}}$ , and performance is strongest between altitudes of 3 and 14 km. Our results also show that ICI observations are sensitive to mean mass diameter values up to  $600~{\rm \mu m}$ , although successful retrievals of up to  $800~{\rm \mu m}$  are observed. To assess the vertical resolution of the retrievals, we computed approximations of averaging kernels on the model predictions. We estimate the resolution of IWC profiles to be  $\sim 2.5~{\rm km}$ . Retrievals of mean mass diameter achieve an estimated resolution of  $2.5~{\rm km}$  at an altitude of  $5~{\rm km}$ , with reduced resolution at higher altitudes.

No operational product currently provides ice mass vertical information derived from passive microwave observations. However, this study demonstrates that ICI can fill this gap thanks to the presence of both microwave and sub-millimetre channels, with the sub-millimetre wavelengths providing particularly high sensitivity to cloud ice. Furthermore, the relatively broad swath of ICI observations lead to a higher spatial and temporal coverage than radar and lidar products can achieve. The global and long-term dataset that ICI will offer could therefore act as a valuable complement to CloudSat or EarthCARE-based retrievals.

Future efforts could explore the inclusion of the Microwave Imager (MWI) observations to improve retrievals at low altitudes — a natural next step given that MWI is to be launched on the same platform as ICI.

#### 1 Introduction

Atmospheric ice plays an important role in the hydrological cycle. Although it constitutes only a small fraction of Earth's total water reservoirs, it has a significant impact on the energy transfer between Earth's surface and the atmosphere. This energy exchange is driven by the release of latent heat upon formation of cloud ice, through precipitation, and through changes in humidity. Ice clouds are also modulators of outgoing longwave radiation, and have been shown to have a net warming effect

on the planet (Matus and L'Ecuyer, 2017; Deutloff et al., 2025). While the global mean ice water path (IWP) has remained relatively stable over time, there have been significant regional changes over recent decades (Pfreundschuh et al., 2025). Due to its effect on atmospheric processes, its radiative impact, and the observed regional changes, a strong understanding of atmospheric ice is essential.

However, our knowledge of atmospheric ice remains limited. This has broad ramifications, with cloud feedbacks named as the greatest uncertainty in equilibrium climate sensitivity estimations (IPCC, 2021). Such uncertainties are partially attributed to a long-standing struggle to represent ice clouds in climate models. Better information on atmospheric ice could help to address this challenge. For example, the mass of ice is strongly related to the cloud radiative forcing. The vertical distribution of ice mass is particularly valuable information. This information helps to characterise the transfer of energy and water, and to estimate the radiative impact at different layers of the atmosphere. The vertical distribution of ice has also been shown to impact radiative heating rates (Mather et al., 2007; Hartmann and Berry, 2017), which drive convective systems and large-scale circulation. Additionally, data on the mass-weighted diameter of ice cloud particles could help to constrain model parameterizations of microphysical cloud processes. Therefore, to increase our confidence in climate model predictions, we need more reliable data on ice mass.

Consistent, global information on atmospheric ice must come from satellites. While numerous satellites measure ice clouds, the sensitivity of the sensors varies. Many satellites provide data on the entire amount of ice mass contained in a vertical column of the atmosphere (Platnick et al., 2003), characterised by the ice water path (IWP). Only a few existing satellite missions provide data on the vertical distribution of ice mass. Active sensors are currently the most reliable sources of vertical information, enabling radar and lidar-based retrievals of variables such as the ice mass per unit volume of air, defined as ice water content (IWC, kg m<sup>-3</sup>) (Deng et al., 2015; Cazenave et al., 2019). EarthCARE is among the newest active satellite missions employing active sensors for atmospheric observations, and will serve as a basis for retrieval products providing high-quality and high-resolution IWC data (Illingworth et al., 2015).

Compared to active sensors, the number of passive sensors capable of retrieving vertical information is limited. In the context of ice cloud observations, passive satellite sensors can be divided into two categories: infrared/optical and microwave. Infrared sensors are generally limited to cloud top information, and thus their measurements are not optimal for deriving vertical information. However, developments in machine learning retrieval methods have enabled successful retrievals of IWC from geostationary infrared measurements (Amell et al., 2024). Meanwhile, microwave radiation is typically able to penetrate through cloud, and therefore does provide information on cloud layers, including the vertical extent. Unlike passive thermal infrared measurements, microwave radiances are not directly related to the physical cloud top temperature, which is needed for long-wave flux estimates, and must be estimated from retrieved vertical information. However, the retrieval of cloud ice information from microwave measurements is associated with another challenge: most microwave sensors measure at too low frequencies to be truly sensitive to the presence of cloud ice. In fact, until 2024 there existed an observational gap between infrared and microwave wavelengths — the sub-millimetre region (Wu et al., 2024). Radiation in this region has a particularly high sensitivity to ice mass, making it critical for the improvement of atmospheric ice observations (Evans et al., 2002).

The observational gap in the sub-millimetre region is now being bridged with the upcoming launch of the Ice Cloud Imager (ICI) and the recently launched Arctic Weather Satellite (AWS, Eriksson et al., 2025). Although AWS is the first to measure at sub-millimetre wavelengths (325 GHz), and will therefore provide ice-sensitive measurements, ICI will be the first operational passive microwave and sub-millimetre sensor specifically designed for the retrieval of ice mass quantities. The ICI radiometer will be hosted on the EUMETSAT (the European Organisation for the Exploitation of Meteorological Satellites) Polar System – Second Generation (EPS-SG) series of satellites. Using observations at frequencies between 183 GHz and 664 GHz, ICI will officially offer retrievals of IWP, mean mass height  $Z_{\rm m}$ , and mean mass diameter  $D_{\rm m}$  (Eriksson et al., 2020).

Although ICI is set to provide a novel dataset of ice mass column variables, the challenge of obtaining vertical information from passive sensors remains. However, ICI will measure across 13 channels, each with varying sensitivity to different atmospheric layers. This implies that ICI observations indirectly contain vertical information. Although active sensor measurements provide this information directly, ICI offers several advantages over current products based on active sensor data. Firstly, ICI will have a combined lifetime of around 22 years, contrasted with EarthCARE's estimated lifetime of 3 years. To examine trends in atmospheric ice, a long-term dataset is desirable. Secondly, ICI will observe with a wider swath than EarthCare. As a result, ICI measurements will contain additional horizontal information, and global coverage will be reached in a shorter timeframe. Therefore, although passive microwave sensors will not provide information at a spatial resolution comparable to active sensors, ICI could act as a complementary source of data to EarthCARE and similar future missions.

Importantly, there does not yet exist a global retrieval product offering vertical profiles derived from passive microwave observations. However, there have been multiple studies demonstrating that it is possible to derive such information. Evans et al. (2012) developed an algorithm to retrieve profiles of IWC and  $D_{\rm m}$  from microwave and sub-millimetre observations, and applied it to flight campaign measurements from the Compact Scanning Submillimeter-wave Imaging Radiometer (CoSSIR). There have also been several studies performed in connection to ICI. Using neural networks, Wang et al. (2016) retrieved vertical mass profiles of individual hydrometeor types from ICI simulations over Europe. Liu et al. (2018) have developed an algorithm for retrievals of both liquid and frozen water contents. Retrieval tests were conducted on simulations of ICI performed along one A-train orbit. The benefits of using sub-millimetre observations for the derivation of vertical information have been explored by Pfreundschuh et al. (2020), where passive-only retrievals were compared with a combined active-passive approach, using simulations that assumed airborne geometry. The retrievals were later validated using real observations in Pfreundschuh et al. (2022).

The aforementioned studies all indicate that ICI's potential extends past column-integrated variables. However, such studies have been limited either in geographic extent or to airborne sensors. Properly assessing ICI's capability requires global retrievals over the full range of possible atmospheric states and meteorological conditions.

In this study, we address the question: how well can vertical information be retrieved from ICI observations? To assess this question, we explore the feasibility of retrieving IWC profiles and profiles of mean mass diameter,  $D_{\rm m}^{\rm IWC}$ , from ICI observations. We train a machine learning model on state-of-the-art ICI simulations, producing probabilistic retrieval estimates. The same simulated observations will be used for the operational ICI level-2 (L2) product offered by EUMETSAT. The simulations were generated as part of a EUMETSAT study, have been validated by May et al. (2024), and have facilitated a characterisation

of retrieval performance for the official ICI L2 variables. By using these same simulations when exploring the retrieval of vertical profiles, we can gain a clearer expectation of the performance of a possible IWC and  $D_{\rm m}^{\rm IWC}$  global retrieval product.

In Sect. 2, we present the ICI sensor, and Sect. 3 presents the simulation framework used to generate the ICI retrieval database. In Sect. 4, the retrieval approach is described. The retrieval results are presented and discussed in Sect. 5, with retrievals of IWC covered in Subsect. 5.1. and retrievals of  $D_{\rm m}^{\rm IWC}$  covered in Subsect. 5.2. Subsection 5.4 contains a discussion of observation information content, and estimations of retrieval resolution are presented in Subsect. 5.5.

## 2 The Ice Cloud Imager

110

ICI will be launched as part of the EPS-SG series of satellites. ICI will be hosted on board the Metop-SG B satellite, which will primarily host microwave instruments intended for imaging. The Metop-SG B satellite will have three successive launches, where each separate satellite has an estimated lifetime of around 7.5 years. In total, the ICI sensors will deliver continuous observations for  $\sim$ 22 years. The launch of the first ICI is currently estimated to be 2026.

The ICI instrument is a conically scanning radiometer, measuring at an incidence angle of  $53\pm2^{\circ}$ . A swath of width  $\sim$ 1700 km is observed. The relatively wide swath width will enable ICI to achieve close to global coverage within a 24-hour period of observations. Further details on the ICI instrument are available in Bergadá et al. (2016) and Eriksson et al. (2020).

ICI will observe using 13 channels. The effective footprint size at 3 dB for all channels is approximately 16 km. The channels span a frequency range of 183 GHz to 664 GHz. Nine channels, centred at frequencies of 183.31, 325.15 and 448.0 GHz, are designed to cover the water vapour absorption lines. These channels will measure at vertical polarisation. The four remaining channels, two each at frequencies of 243.2 and 664.0 GHz, will measure at both vertical and horizontal polarisation. Due to reduced atmospheric attenuation at these frequencies, these channels will allow for observations at lower altitudes. The variation in transmission between channels and across a range of atmospheric scenarios is shown in Fig. 1. As a result of the wide range of frequencies covered by the channels, and therefore the varying degrees of attenuation experienced at each of the frequencies, ICI observations will be sensitive to ice mass located at multiple atmospheric layers.

Also onboard the Metop-SG B satellite will be the Microwave Imager (MWI) (Lupi et al., 2016). Like ICI, MWI is a conically scanning radiometer measuring at an incidence angle of 53°. It will observe across frequencies ranging from 18.7 GHz to 183 GHz, allowing for higher sensitivity to precipitation.

## 3 Retrieval approach

## 3.1 Retrieval theory

Retrieving geophysical variables from remote sensing observations can be challenging, especially when the inverse problem is non-linear and ill-posed, as is the case for ICI. For an ill-posed problem, a single observation can map onto multiple descriptions of the atmospheric state. Retrieving a single estimate of a geophysical quantity is therefore unrealistic in this context. It is more

**Figure 1.** Transmission to space of ICI channels as a function of altitude. The shaded regions indicate the range of transmission values across various atmospheric scenarios, where the upper bound corresponds to a tropical atmosphere and the lower bound corresponds to a subarctic winter atmosphere. The dotted line represents an optical thickness of 1. Channels are shown in the legend according to their frequency in GHz.

suitable to approach the problem from a probabilistic standpoint, and retrieve the posterior probability density function (PDF) of the desired variable.

There are multiple retrieval methods that take such an approach, such as the optimal estimation method (OEM, Rodgers, 2000) or Bayesian Monte Carlo Integration (BMCI), as used in Evans et al. (2002). However, the drawback of both these methods is the assumption of Gaussian statistics. In the case of OEM, both the uncertainties and the retrieved distribution are assumed to follow a Gaussian distribution. In BMCI, the retrieved distribution may take any form, but the observational error is still assumed to follow a Gaussian distribution. Both methods have been used to great success in a range of remote sensing retrievals. In fact, the operational ICI retrievals at EUMETSAT will use BMCI.



Machine learning offers an alternative approach where neither assumptions on the retrieval distribution nor the error distributions are required. However, many machine learning approaches provide only a single estimate of a quantity, without any associated uncertainty. To better suit the ill-posed nature of our inverse problem, a quantile regression neural network (QRNN) can be used. As described in Pfreundschuh et al. (2018), QRNNs minimise a quantile loss function. The result is the prediction of a sequence of quantiles, allowing for the estimation of a discrete cumulative distribution function (CDF). The retrieved CDF can then be easily transformed into a PDF. The retrieved PDF or CDF allows for representation of the uncertainties associated with the retrieval. By extension, if the training data are distributed as the a priori, then the obtained PDF or CDF describes the a posteriori knowledge.

Pfreundschuh et al. (2018) were the first to apply QRNNs to atmospheric remote sensing retrievals. They performed a comparison of QRNN and BMCI, retrieving column water vapour from simulated passive microwave observations, and cloud top pressure from MODIS (The Moderate Resolution Imaging Spectroradiometer, Platnick et al., 2003) observations. The study demonstrated that QRNN achieves accuracy comparable to BMCI, both in terms of predictions and of estimates of a posteriori distributions. In fact, QRNN outperforms BMCI when fewer data are available, indicating that a smaller database suffices for QRNN. Additionally, they estimated that QRNN can perform retrievals at least an order of magnitude faster than BMCI.

The quantile regression approach has also been applied to the retrieval of ice mass profiles. Amell et al. (2024) retrieved IWP and IWC with a convolutional neural network (CNN) using quantile regression. The retrievals were extensively validated and shown to perform well. While the retrieval model used in Amell et al. (2024) leverages horizontal spatial information, it does not require explicit a priori assumptions on vertical correlations beyond those implicitly captured in the reference data. This contrasts with Pfreundschuh et al. (2020, 2022), where OEM-based retrievals of IWC relied on the assumption of exponentially decaying vertical correlations.

## 3.2 Retrieval model implementation







The retrievals performed in this study use a QRNN to predict quantiles of column integrated ice mass quantities (IWP,  $D_{\rm m}$ , and  $Z_{\rm m}$ ) and ice mass profiles (IWC and profiles of mean mass diameter  $D_{\rm m}^{\rm IWC}$ ). Each layer of a profile quantities is predicted independently, i.e. it is treated as a separate output of the QRNN. The inputs to the retrieval model are antenna temperatures  $T_{\rm a}$  for all 13 ICI channels, and ancillary data in the form of surface type, surface temperature, and surface pressure. Following the approach taken in May et al. (2024), measurement noise was added to all simulated  $T_{\rm a}$ , using 75% of the NE $\Delta$ T estimates provided in Eriksson et al. (2020). Each noise value was generated using a random Gaussian noise generator with standard deviation of  $0.75{\rm NE}\Delta$ T. A new noise value was randomly generated each time a training sample passes through the network, serving as data augmentation. Further details on the neural network architecture are given in Appendix A.

We select QRNN for our retrievals of IWC from ICI observations based on its demonstrated advantages. Pfreundschuh et al. (2018) have highlighted the benefits of QRNNs — providing computational efficiency while maintaining the accuracy and uncertainty estimation capabilities of BMCI. Given these benefits, along with the successful use of QRNNs for retrieving column ice mass variables from ICI observations (May et al., 2024), QRNNs can serve as a replacement for BMCI in ICI retrievals. Additionally, the flexibility of QRNNs suits the retrieval of ice mass profiles, since each profile level can be retrieved independently, negating the need for vertical correlation assumptions. Only a single neural network is used, without additional pre- or post-processing steps, which is a relatively straightforward approach. While studies such as Wang et al. (2016) have successfully taken a more advanced approach, designing a retrieval algorithm for ICI based on multiple neural networks, our approach aims to simplify the analysis of retrieval performance. As a result, focus is directed to the potential of ICI data, avoiding the need to disentangle any possible impacts of a more complex approach.

The retrieval model allows for the estimation of PDFs for each output variable. However, it remains useful to represent the retrieval with a single estimate. A straightforward approach is to use either the mean or the median of the PDF. Alternatively, a random sample of the PDF can act as the single estimate. This latter approach is generally only useful when computing aggregated statistics, and less so when the aim is to look at individual cases. In the results shown in this study, the mean of each retrieved PDF is taken as the single estimate.

The retrievals performed in this study will inevitably have some degree of error. Potential errors in the predictions largely stem from two sources. The first type of error is the epistemic uncertainty. This uncertainty is associated with how well constructed the model is, and can originate from a lack of training data or poorly chosen model hyperparameters. As a result, the epistemic uncertainty can be interpreted as potentially avoidable, or reducible, error. This error is not modelled by QRNNs,

and alternate approaches must be taken to estimate it, although this is challenging. The main consideration to mitigate this error is to include a large amount of data, which is a key motivation behind the high number of simulations within our retrieval database.

The second error source is the aleatoric uncertainty. This arises from the nature of the ill-posed problem and is a reflection of the difficulty in determining the atmospheric state from the observations. Although this error is irreducible, it can be represented in the training data and, by extension, the retrieved PDF. We reflect the natural variability of the atmospheric state through the randomisation of variables such as surface emissivity and particle model, as discussed in Sect. 4.2 and validated in May et al. (2024). Uncertainties associated with the model inputs will also propagate through to the aleatoric uncertainty. Measurement noise is added to all simulated observations during training and is therefore represented in the CDF. The same approach could be taken for the ancillary input data, such as the ERA5 surface characteristics (see Sect. 4.2), but this requires good error characterisation and is therefore more challenging. Therefore, our retrieved CDFs represent the majority, but not all, of the aleatoric uncertainty associated with the retrieval, where this uncertainty is dominated by natural variability and observation noise. Given the accuracy of present day machine learning algorithms, the aleatoric uncertainty captured in the CDF dominates the total uncertainty.

## 4 ICI retrieval products






In order to perform inversions of ICI observations, the retrieval model used in this study must be trained on a retrieval database, i.e. a dataset of ICI observations and associated ice mass products. The challenge in creating a retrieval database lies in the fact that, at the time of finalising the ICI retrieval algorithm, there were no operational sub-millimetre missions measuring atmospheric ice. Therefore, no real observations exist that could facilitate the creation of an empirically-based retrieval database. A database of simulated observations is therefore a necessity.

## 4.1 Operational level-2 product

EUMETSAT will offer a L2 product containing ICI retrievals: MWI-ICI-L2. Although the L2 product contains retrievals based on both ICI and MWI observations, retrievals are performed separately for each instrument. ICI observations will be used to retrieve IWP, mean mass diameter  $D_{\rm m}$ , and mean mass height  $Z_{\rm m}$ .  $D_{\rm m}$  and  $Z_{\rm m}$  are conditional on the presence of ice (IWP > 0). The primary output of the MWI-based retrievals is the liquid water path (LWP) (Mattioli et al., 2019).

Details of the retrieval algorithm on the ICI side are provided by the EUMETSAT Satellite Application Facility (SAF) supporting nowcasting (NWC-SAF) within the algorithm theoretical basis document (ATBD), found at Rydberg (2018). Within the algorithm, several pre-processing steps are taken prior to the inversion, such as the remapping of Level-1b (L1b) data and a bias correction scheme. Such steps were developed inside a EUMETSAT study and results are presented in detail in Eriksson et al. (2020).

## 4.2 Retrieval database








High-quality data are a requirement to perform accurate and reliable inversions. A framework was therefore developed with the aim to produce state-of-the-art all-sky simulations. The simulations were generated as part of a EUMETSAT study, in order to produce a retrieval database for use in operational L2 retrievals at EUMETSAT.

An additional requirement is that the number of simulated observations is high enough to statistically represent the true variability. To ensure a robust inversion model, it is desirable that one observation matches multiple states. A high number of simulations helps to achieve this outcome. In light of these requirements, the design of the framework needed to balance computational efficiency with the need for detailed radiative transfer calculations. This section provides an overview of the inputs, outputs, and process of the database generation framework. For further details, an in-depth description of the simulation environment is presented in May et al. (2024). This includes the multiple considerations taken to meet the above requirements. A total of approximately 9.4 million cases are simulated. A single case represents observations for each of the 13 channels, with all observations remapped to the same on-ground footprint.

The process of generating the database began with the generation of three-dimensional atmospheric states. The use of threedimensional states was motivated by the need for information in both the along- and across-track direction, which allows for the inclusion of the two-dimensional antenna response later in the scheme. To ensure that the database reflects global variability, the coverage corresponds to CloudSat (Stephens et al., 2002) overpasses during the years 2009 and 2010. CloudSat overpasses offer only two-dimensional coverage: height and along-track. To expand the coverage into three-dimensional space, the algorithm of Barker et al. (2011) was implemented. This requires additional data of two-dimensional coverage in the alongand across-track directions. MODIS was chosen to fulfil this need, due to its presence on the NASA A-Train constellation and therefore the possibility to colocate observations with CloudSat. Within the Barker et al. (2011) scheme, the colocated MODIS observations are used to extend CloudSat radar reflectivities into the across-track direction. The radar reflectivities are converted to IWC using a look-up table, according to the method described in Ekelund et al. (2020). The decision to construct fields of radar reflectivities, rather than to simply use the IWC data available in CloudSat retrieval products, was taken to avoid incorporating retrieved data from other sources into the database. Furthermore, we could ensure consistency between the microphysical assumptions used in the computation of IWC and the assumptions used during the radiative transfer calculations. The resulting three-dimensional field of IWC is supplemented with ancillary data from ERA5 (Hersbach et al., 2020); this data consists of atmospheric gas quantities, weather conditions, surface classifications, and liquid water content (LWC) profiles. Approximately  $5 \times 10^4$  three-dimensional fields were generated, each measuring approximately 2000 km in the along-track direction and 50 km in the across-track direction relative to the CloudSat sub-satellite path. The fields are henceforth referred to as atmospheric scenes.

Both the conversion of radar reflectivities to IWC and the subsequent radiative transfer simulations are sensitive to the characterisation of ice hydrometeors. We defined six distinct particle models for use in the simulations. Each particle model consists of a habit and particle size distribution (PSD). The habits were selected as a representative sample from the ARTS single scattering database (Eriksson et al., 2018). Table 2 in May et al. (2024) provides an overview of the six particle models,

including the choice of habit and the PSD used for each model. One of the particle models is selected to be used in simulations within one atmospheric scene. Not all particle models are applied equally often. Instead, they are randomly chosen according to a pre-defined probability, which varies between particle model. The choice of the probabilities is motivated in Sect. 3.3 of May et al. (2024).







Due to the presence of vertically and horizontally polarised channels in ICI, it is necessary to consider the orientation of ice particles. Typically, totally random orientation (TRO) is assumed when simulating at microwave frequencies. However, this approach tends to significantly underestimate polarisation differences. To better reflect reality, the effects of orientation were mimicked using an approximation of azimuthally random orientation (aARO) following a scheme developed by Barlakas et al. (2021) and later expanded upon by Kaur et al. (2022). The scheme applies a scaling factor to the extinction values arising from a TRO assumption. The range of scaling factors for each particle models are given in Table 2 in May et al. (2024). The scaling factors were chosen according to findings in Kaur et al. (2022), but increased slightly to account for ICI's sub-millimetre frequency channels potentially producing higher polarisation differences.

Radiative transfer simulations are then performed within the three-dimensional atmospheric scenes. The Atmospheric Radiative Transfer Simulator (ARTS) (Buehler et al., 2025) is used to perform the calculations, producing monochromatic pencil beam brightness temperatures as the output. To perform the forward model calculations, ARTS requires the input of gas absorption models, with the specific sources of each absorption model given in Sect. 3.4 of May et al. (2024).

At the frequencies that ICI will measure at, the contribution of the surface to the calculations can be significant. Two surface emissivity models are used: Tool to Estimate Sea-Surface Emissivity from Microwaves to sub-millimetre waves (TESSEM<sup>2</sup>) (Prigent et al., 2017) and Tool to Estimate Land-Surface Emissivities at Microwave Frequencies (TELSEM<sup>2</sup>) (Wang et al., 2017). However, TELSEM<sup>2</sup> has certain limitations at high frequencies, as a constant emissivity is assumed above 183 GHz (and as low as 85 GHz for some surface types). To account for this limitation, a probabilistic model was developed for snow and sea ice surfaces. Within the model, emissivities are sampled from a multivariate Gaussian distribution, where the distributions are empirically derived, based on emissivities from Hewison et al. (2002); Harlow (2009); Harlow and Essery (2012); Munchak et al. (2020).

Radiative transfer simulations were performed using the DISORT (Discrete Ordinate Radiative Transfer) scattering solver. The reader is directed to Sect. 3.4 of May et al. (2024) for details of the simulations performed. The application of the spectral response function is described in Sect. 3.5 of May et al. (2024). The spectral radiances obtained from the all-sky simulations are subsequently converted to brightness temperatures using the inversion of the Planck function. The resulting brightness temperatures span both the along- and across-track directions of the scene. However, if a single pencil beam calculation is used to represent an ICI observation, there will be an overestimation of the decrease in radiance caused by the presence of ice (Barlakas and Eriksson, 2020). To avoid this, and thus capture the beam-filling effect (Davis et al., 2007), the two-dimensional field of brightness temperatures  $T_{\rm b}$  was integrated over the ICI sensor field of view:

$$T_{\rm a} = \int_{\Omega} T_{\rm b}(\Omega) G(\Omega) d\Omega. \tag{1}$$

 $\Omega$  is the solid angle and  $G(\Omega)$  is the normalised antenna gain function provided by EUMETSAT.  $T_{\rm a}$  is the observed antenna temperature. As a result of the inclusion of the two-dimensional antenna pattern, each  $T_{\rm a}$  takes information from the whole ICI footprint and therefore includes horizontal information. All channels are assumed to share the same ground-level footprint as the 183 GHz channel, ensuring that the final  $T_{\rm a}$  data are consistent with the remapping to be applied in the operational L2 algorithm.

## 4.3 Database for research purposes




In the course of generating the retrieval database presented in May et al. (2024), we also created an additional 'extended' database for research purposes by storing, or reconstructing, variables used during the simulations. This includes the variables of interest for this study: vertical profiles of IWC and mean mass diameter,  $D_{\rm m}^{\rm IWC}$ . We note that the definition of IWC used in this study includes all ice mass in a vertical column, i.e. both cloud ice and precipitating ice. By preserving this information, the extended database is a useful resource for the exploration of the further retrieval possibilities of ICI.

The input data to the simulations are not given on the same altitude grid for each scene. In order to later retrieve IWC at consistent altitude intervals, it was necessary to remap the original IWC data onto the same grid. Each vector of IWC was linearly interpolated onto a fine grid. The interpolated data was then averaged within 500 m bins. The result is 40 IWC data points, equally spaced between 0 km and 20 km, for each simulation case. The same approach was also taken for temperature profiles. The retrieval model is trained on the 500 m vertical resolution IWC data, and the retrieved profiles maintain this same resolution. In other words, the retrieval model includes 40 individual IWC values as part of its output for each input observation.

The mean mass diameter,  $D_{\rm m}^{\rm IWC}$ , is defined as the ratio of the fourth and the third moments of the PSD, as per Delanoë et al. (2014):

$$D_{\rm m}^{\rm IWC} = \frac{\int_0^\infty N(d_{\rm eq}) d_{\rm eq}^4 dd_{\rm eq}}{\int_0^\infty N(d_{\rm eq}) d_{\rm eq}^3 dd_{\rm eq}}.$$
 (2)

In the above equation,  $d_{\rm eq}$  refers to the equivalent volume diameter, i.e. the diameter of a sphere of ice with the same mass as the particle.  $N(d_{\rm eq})$  is the PSD. It is noted that the  $D_{\rm m}$  provided in the L2 product, and retrieved in May et al. (2024), is the column integrated mean mass diameter.  $D_{\rm m}^{\rm IWC}$  in Eq. 2 is a vertical profile of mean mass diameter, and is the focus of this study.

Although profiles of  $D_{\rm m}^{\rm IWC}$  were not stored during the generation of the retrieval database, it is possible to reconstruct the PSD. This requires IWC and temperature profiles stored in the database, and knowledge of the particle model used for each simulation. While this information is available, the IWC data stored during database generation are antenna weighted values. As such, our calculations of  $D_{\rm m}^{\rm IWC}$  are somewhat approximate, though not to a large extent and will likely have a very minimal impact.  $D_{\rm m}^{\rm IWC}$  profiles were constructed and are retrieved at the same 500 m resolution as the IWC profiles.

## 310 5 Results and discussion





#### 5.1 Retrieval of ice water content

The first step in evaluating the retrievals is to compare the retrieved values with the 'true', or target, values across the range of IWC. The mean of each retrieved IWC CDF is taken as the single retrieval estimate, and compared against its corresponding simulated database case.

Overall retrieval performance, as shown in Fig. 2, is evaluated in two ways. Firstly, we compute the mean, median, and quantiles across the set of retrieved CDF means. These statistics are shown as a function of the true value. Secondly, to provide an overall metric of the retrieval accuracy across the range of possible values, we calculate the median fraction error (MFE) of the retrievals. Note that this is a metric applied post-retrieval, and is not used as a neural network loss function. The MFE accounts for the orders of magnitude spanned by IWC, and is defined by Brath et al. (2019) as

320 
$$MFE(x) = \text{median}\left(\exp_{10}\left(\left|\log_{10}\frac{x_{\text{retrieval}}}{x_{\text{true}}}\right|\right) - 1\right).$$
 (3)

Retrieval performance was examined at individual atmospheric layers. Figure 2 presents a representative sample of the layers, at altitudes 3.25 km, 7.25 km, and 9.25 km. At 3.25 km (Fig. 2a), the mean and median follow the identity line between  $10^{-2} \leq IWC \leq \sim 3 \times 10^{-1} \mathrm{g m}^{-3}$ , but display decreased sensitivity to the true IWC as IWC increases. Retrieval variability, i.e. the spread between the 16% and 84% quantiles, is highest for IWC  $< 10^{-2} \mathrm{g m}^{-3}$ . The same trends are visible in the MFE.

Performance improves when increasing in altitude to 7.25 km (Fig. 2b) and, to an even greater extent, to 11.25 km (Fig. 2c). The most notable differences are the lower variabilities achieved across the entire IWC range. This is reflected in a lower MFE, except at very low IWC. In fact, at 11.25 km, the median and the mean remain sensitive to the true IWC to as low as 1 mg m $^{-3}$ . At high IWC, the improvement in performance is the most significant. Both the mean and the median now follow the identity line up to the maximum IWC, around 5 g m $^{-3}$  for 7.25 km and 3 g m $^{-3}$  for 11.25 km. In other words, at 7.25 km and 11.25 km, retrievals are sensitive to IWC an order of magnitude greater than at 3.25 km.

Other altitudes were also examined (not shown here), with similar trends observed. Performance decreased further at altitudes lower than 3.25 km, where overall variability increased and less sensitivity to the true IWC was observed at both low and high IWC. Performance also began to decrease above 14 km. In these high-altitude cases, significant overestimation at high IWC was seen, corresponding to MFE  $\geq$  400 %. This suggests that our IWC retrievals are relatively inaccurate at altitudes greater than 14 km.

The variation of performance with altitude is not an unexpected result, and there are several factors that may contribute to the poorer retrieval performance at some altitudes. For higher atmospheric layers, uncertainty may arise from a struggle to constrain the altitude from the observations. In order to constrain the altitude of ice, a channel must also experience water vapour absorption at the same altitude. However, due to the low amount of water vapour at high altitudes  $> 14 \,\mathrm{km}$ , there is little variation in the transmitted signal and the altitude information is lost. Additionally, the PSDs used in the database simulations have relatively small ice crystals at low temperatures. Therefore, at the low temperatures experienced at high altitudes, the impact on ICI radiances will be relatively weak due to the implemented PSDs.

At lower altitudes, signal attenuation and surface interference will lessen the impact of ice on ICI's signal and thus degrade performance. The influence of attenuation in clear-sky conditions varies between channels, as shown in Fig. 1. The degradation in retrieval performance seen at 3.25 km can therefore be understood in terms of the number of channels actively providing information. In clear-sky conditions, the higher frequency channels (448 GHz and 664 GHz) exhibit almost no transmission around 3 km, regardless of the atmospheric scenario. Additionally, transmission in the 183 GHz and 325 GHz channels will be very low for a significant number of the retrieval cases. Even the presence of a thin cloud will lead to a substantial further increase in attenuation, particularly at 664 GHz. Although the 664 GHz channel is less affected by atmospheric absorption than the 448 GHz channel — allowing it to sense lower altitudes in clear-sky humid conditions — its higher frequency results in greater scattering by cloud ice crystals. As a result, the 664 GHz channel will experience a loss of signal at lower altitudes than the 448 GHz channel in cloudy conditions, degrading the performance of the IWC retrieval.







At 7.25 km, the number of active channels is higher, corresponding with the improved performance relative to the lower altitudes. However, in some atmospheric scenarios, the water vapour channels will experience relatively low transmission. Attenuation will be particularly high at 448 GHz. Increasing in altitude to 11.25 km improves retrieval performance further, corroborated by the fact that all channels display high transmission at this altitude.

Although the 243 GHz channel remains active at low altitudes, this also means that its measurements are sensitive to the surface. This surface interaction can obscure the presence of cloud ice in the measured signal and lead to challenges for the retrieval model. There may also be an impact from uncertainties in the modelling of emissivities. May et al. (2024) highlighted that ICI IWP retrievals tend to perform worse at higher latitudes. This result was partially attributed to the presence of snow and sea-ice surfaces, since the emissivities of such surfaces remain poorly modelled. Since most of the ice mass lies at lower altitudes in high-latitude regions, there could be an intertwined effect of low-altitude ice clouds above snow and sea-ice, which in turn contributes to higher variability in the retrievals.

As a result of the aspects discussed above, we can deduce that the measurements obtain a vertical resolution varying with altitude. This is explored further in Sect. 5.4 by examining the structure of the averaging kernels. For IWC, the vertical resolution is shown to be  $\sim$ 2.5 km.

To further evaluate the performance of the IWC retrievals, individual scenes were examined. Examples of these individual scenes are presented in Fig. 3 and Fig. 7. Overall, retrievals of IWC agree with the database in terms of magnitude. In most scenes examined, both high and low IWC appear to be accurately retrieved within the range of 1 mg m<sup>-3</sup> < IWC < 1 g m<sup>-3</sup>. Some underestimation does occur within the cloud structures, but this generally occurs for regions of lower-IWC (e.g. at  $-14^{\circ}$  latitude in Fig. 7), or for finer cloud features (e.g. at  $-21^{\circ}$  latitude in Fig. 7). These results are consistent with the retrieval performance discussed previously.

Larger scale cloud structures are clearly present in both the retrievals and the database. The overall cloud structures appear to be more vertically diffuse in the retrievals. This diffusivity is especially visible in panels (c) of Fig. 3 and Fig. 7, where IWC is overestimated both above and below the clouds. It should be noted that this difference is calculated as relative to the true IWC values, which amplifies the difference if the true IWC is very small. Nonetheless, this is further evidence that the effective resolution of the retrieval model (~2.5 km) is coarser than the retrieval resolution of 500 m, explored further in

**Figure 2.** IWC retrieval performance for altitudes of 3.25 km, 7.25 km, and 11.25 km is shown in panels (a), (b), and (c), respectively. The mean of the retrieved CDF is taken as a point estimate, and the mean, median, 16th quantile, and 84th quantile of the retrieved distribution mean are plotted as a function of the true value. The bias and the correlation coefficient r are calculated for all true cases with IWC >  $10^{-3}$  g m<sup>-2</sup>, taking the retrieval mean as the single estimate.

Sect. 5.4. Several smaller features are captured in both the retrievals and the database, and those not captured in the retrievals appear to be cases with relatively low IWC. However, the retrievals sometimes produce ubiquitous low-level ice clouds with an IWC of  $\sim 0.01~\rm g~m^{-3}$ , such as in Fig. 3. These low-level clouds appear as regions of high overestimation in panel (c) of Fig. 3. However, they do not appear in Fig. 7. Such clouds were not found to be a general feature in all scenes examined, but did occasionally appear. Although no systematic investigation was carried out, we found that rain was present underneath the low-level cloud retrievals for the case shown in Fig. 3, which may explain their occurrence. The occurrence of these low-level clouds is explained by retrievals at low altitudes tending towards an a priori distribution as IWC decreases, as shown in Fig. 2b. As a result, occasional retrievals of up to  $0.01~\rm g~m^{-3}$  do occur, even when the true IWC is significantly lower.





Also observed in Fig. 3, Fig. 7, and other examined scenes is a 'striping' effect. This arises because the model is trained to retrieve each profile individually. As a result, the retrievals ignore correlations between neighbouring profiles, which are present in reality. The striping is therefore a statistical artefact due to fluctuations in noise between profiles. In Fig. 3d, the retrieval is performed on noise-free radiances. The striping effect is no longer present, illustrating that instrument noise does have an impact on the retrievals. The influence of noise is stronger at low IWC, where the cloud signal is only slightly higher than the noise. A less noisy instrument would therefore improve the detection of IWP and IWC in thin clouds, where the cloud signal is of comparable magnitude to the noise. However, we note that the retrievals shown in Fig. 3b and Fig. 7b are represented by the mean of the retrieved CDF. Therefore, even in the presence of striping, it is highly likely that the 'true' values are contained in the full uncertainty estimation provided by the QRNN.

The striping effect is also observed in Liu and Adams (2024), when performing nadir-only IWC retrievals from submillimetre observations. The effect was subsequently removed after the inclusion of multi-angle observations within their algorithm. Although ICI will not offer multi-angle observations, and our study focuses on an assessment of ICI's capabilities rather than algorithm development, the results of Liu and Adams (2024) suggest potential for future developments of an ICI IWC product. For example, knowledge of neighbouring observations could be incorporated into the retrieval. The flexibility of a machine learning approach offers an advantage in this case. One avenue could involve providing the neural network with  $T_{\rm a}$  from neighbouring swath positions, allowing the model to adjust its prediction accordingly.






Making use of the retrieved quantiles can also provide insight into retrieval performance. In panel (e) of Fig. 3, we show the spread between the 1st and 99th quantiles relative to the distribution's mean, considering only data points with a mean > 1 mg m<sup>-3</sup>. The spread represents the uncertainty associated with the retrieval, as discussed in Sect. 3.2. The most problematic cases in this scene, such as the ubiquitous low-level clouds, are associated with the largest uncertainties. The high uncertainties here suggest a potential technique for flagging and rejecting such cases by an end-user, and they highlight the value of a retrieval model that provides uncertainties. However, we stress that the removal of high-uncertainty cases would impact overall statistics.

Many IWC scenes were examined. The scenes consistently showed the same trends described above: general agreement in cloud structure and IWC magnitude, slight vertical diffusivity, and the striping artefacts.

In Fig. 4 and Fig. 5, one-dimensional distributions and two-dimensional zonal means of IWC are shown. The comparison between database and retrieval distributions allows us to evaluate whether there is a systematic over- or under-estimation of IWC in a particular IWC range, latitudinal region, or layer of the atmosphere. Furthermore, the retrievals are only useful if both they and the database cases are statistically consistent with reality. In the absence of a ground truth to compare to, the same distributions were calculated using IWC data from the DARDAR (raDAR/liDAR) 3.1 product (Cazenave et al., 2019).

In Fig. 4a, the overall distribution of database IWC is presented for several altitudes. The corresponding distribution for retrieved IWC is shown in Fig. 4b. For IWC < 1 g m<sup>-3</sup>, the distribution of the retrievals exhibit good agreement with the database distributions for all altitudes shown. As expected, agreement between retrievals and database varies with altitude. The highest altitude shown (15.25 km) shows the weakest agreement with the database, most notably at the higher and lower ends of the IWC range.

The region of greatest disagreement occurs around the highest IWC (around 5 g m<sup>-3</sup>), where the retrievals extend to slightly higher IWC than the database. This outcome is somewhat expected since the probability density of IWC in this region is very low and, as previously discussed, fewer cases corresponds to a less constrained model. Retrieving cases of high IWC that are absent in the database is not ideal, since this is an extrapolation outside the variability covered by the database. However, the probability density within this region is so low that the disagreement appears to originate from just several cases of overestimation. In contrast, the overall negative bias observed at high IWC in Fig. 2 indicates that such unrealistic overestimations are rare, and underestimations are far more common.

When comparing both retrievals and database to DARDAR in Fig. 4c, generally good agreement is observed up to 1 g m<sup>-3</sup>. There do exist more low-IWC cases in the database than in the DARDAR product. However, despite the fact that the ICI retrieval database and DARDAR both derive IWC from CloudSat overpasses, there are differences in the retrieval schemes that will produce differences. As discussed in May et al. (2024), DARDAR may fail to identify particularly high cases of ice mass (Bolot et al., 2023). DARDAR also demonstrates high variability between product versions (Cazenave et al., 2019). Pfreundschuh et al. (2025) showed that significant differences even exist between DARDAR and 2C-ICE (Deng et al., 2015).

Figure 3. A comparison of IWC profiles, with database IWC shown in (a), and the corresponding retrieved IWC shown in (b). The retrieved IWC is plotted using the retrieval mean as the single estimate for each case. Panel (c) shows the difference between the retrieval and the database, relative to the database value. Only data points with either database or with retrieved IWC  $\geq 1 \text{ mg m}^{-3}$  are shown. Database IWC with zero values were replaced with 1  $\mu g m^{-3}$  to allow calculation of relative differences. Panel (d) shows a retrieval performed using noise-free radiances as input. Panel (e) shows the difference between the retrieved 0.99 quantile and the 0.01 quantile, relative to the retrieved mean. The scene corresponds to a CloudSat overpass on 28 January 2009 at approximately 17:30.

2C-ICE is an equivalent product that, like DARDAR, also derives information from CloudSat and CALIPSO (Cloud-Aerosol Lidar and Infrared Pathfinder Satellite Observations). Therefore, the differences observed between our data and DARDAR are not necessarily a cause for concern.


Zonal means of IWC from DARDAR, the database, and the retrievals are presented in Fig. 5. Good agreement is seen between database and retrieved IWC. The zonal means are also generally consistent with DARDAR. Expected features, such as the tendency for ice mass to be located at higher altitudes nearer the equator, are visibly present in all three datasets. It is noted that the database and retrieval zonal means appear noisier than DARDAR. However, fewer cases are used in the calculation of the database zonal mean than for the DARDAR zonal mean, and even fewer for the retrieval zonal mean.

**Figure 4.** Distribution of IWC. Panel (a) shows the distribution of IWC in the retrieval database. Not all altitudes are represented. Instead, every fifth altitude is shown, where each altitude corresponds to a 500 m layer. In panel (b), the distribution of retrieved IWC is shown, corresponding to the same layers as shown in panel (a). The overall distribution of all IWC cases in the database and the corresponding retrieved values are shown in panel (c). A distribution of IWC calculated from the DARDAR product from the year 2010 is shown for comparison.

## 5.2 Evaluation of column integrated ice water content

It is already established that ICI can provide reliable estimates of IWP (Eriksson et al., 2020; May et al., 2024). When exploring the potential of using ICI observations to retrieve IWC, the extent to which an integrated IWC profile retrieval accurately represents IWP can therefore serve as an indication of the IWC retrieval's success.

Four IWP datasets are available to compare: the database IWP, retrieved IWP, the vertical integral of database IWC, and the vertical integral of retrieved IWC. The latter two datasets are henceforth referred to as  $\int$ IWC. Database  $\int$ IWC and database IWP are unlikely to be exactly equal. During generation of the database, IWC was calculated on a finer vertical grid so to

**Figure 5.** Zonal mean of IWC present in DARDAR (panel a), in the retrieval database (panel b), and the retrieved cases (panel c). IWC is taken from the DARDAR 3.1 product from the year 2010.

achieve accurate IWP values. However, as discussed in Sect. 4.3, the IWC was then interpolated on to a 500 m grid. Database  $\int$ IWC is calculated using the 500 m resolution IWC. This likely leads to small differences between database IWP and database  $\int$ IWC. Since the retrievals are trained on the 500 m resolution IWC data, database  $\int$ IWC should be used as the best point of comparison.



Retrieved IWP and retrieved  $\int$  IWC are plotted as a function of database IWP in Fig. 6a. Good agreement is seen between retrieved IWP and retrieved  $\int$  IWC for IWP  $\geq 10~{\rm g~m^{-2}}$ . Overall distributions of IWP from the four alternative IWP datasets are presented in Fig. 6b. Strong agreement is seen between the four datasets for IWP  $> 0.01~{\rm kg~m^{-2}}$ . Notably, at high IWP, retrieved  $\int$  IWC aligns more closely with the database IWP than the retrieved IWP does.

The distribution of retrieved  $\int$  IWC does exhibit disagreement at the lowest IWP values, characterised by a peak in density at IWP  $\sim 3~{\rm g~m^{-2}}$ , followed by a sharp decline. This feature arises from the preprocessing of IWC data before training the retrieval model. In reality, low IWP cases are dominated by near-zero IWC cases. However, prior to training, cases of IWC  $

Figure 6. Comparison of IWP calculated from retrieved IWC to database IWP. In panel (a), the mean, median, 16th and 84th percentiles are shown. Panel (b) shows the overall distribution of IWP obtained in four different ways: Database IWP, retrieved IWP, the column integral of database IWC, and the column integral of retrieved IWC. The distribution of IWC from the DARDAR 3.1 product is also shown in panel (b). Panel (c) shows  $Z_{\rm m}$  calculated from database IWC and from retrieved IWC, with a direct retrieval of  $Z_{\rm m}$  shown as comparison. Panel (d) shows  $D_{\rm m}$  calculated from database IWC and  $D_{\rm m}^{\rm IWC}$ ,  $D_{\rm m}$  calculated from retrieved IWC and  $D_{\rm m}^{\rm IWC}$ , and a direct retrieval of  $D_{\rm m}$ . The legend in panel (d) applies to both panel (c) and (d). Distributions of  $Z_{\rm m}$  and  $D_{\rm m}$  are calculated only for cases corresponding to IWP >  $10^{-2}~{\rm kg~m}^{-2}$ 

As a secondary test, we also compare the column variable  $Z_{\rm m}$  in panel (c) of Fig. 6, which can be calculated from IWC according to Eriksson et al. (2020). Three distributions are shown: database-derived values, retrieval-derived values, and a direct retrieval of the column variable. Deriving  $Z_{\rm m}$  from retrieved IWC is consistent with  $Z_{\rm m}$  derived from database IWC. Differences occur mainly at the lowest and highest altitudes, in line with our conclusion that IWC can be most accurately estimated between altitudes of 3 and 14 km. The frequency of these cases is, however, very low. Direct retrievals of  $Z_{\rm m}$  exhibit the same limitations at the highest and lowest altitudes.

## 5.3 Retrieval of Dm profiles

Retrieval performance for  $D_{\rm m}^{\rm IWC}$  at altitudes of 3.25 km, 7.25 km, and 11.25 km is presented in Fig. 8. As a metric of retrieval performance, we use the root mean square error (RMSE), since  $D_{\rm m}^{\rm IWC}$  does not vary over orders of magnitude as IWC does. At 3.25 km (Fig. 8a) there is a negative bias present across almost the entire range of  $D_{\rm m}^{\rm IWC}$ , except for extremely low  $D_{\rm m}^{\rm IWC}$ 

Figure 7. A comparison of IWC profiles, with database IWC shown in (a), and the corresponding retrieved IWC shown in (b). The retrieved IWC is plotted using the retrieval mean as the single estimate for each case. Panel (c) shows the difference between the retrieval and the database, relative to the database value. Only data points with either database or with retrieved IWC  $\geq 1 \text{ mg m}^{-3}$  are shown. Database IWC with zero values were replaced with 0.001 mg m<sup>-3</sup> to allow calculation of relative differences. Panel (d) shows IWP across the scene. 'IWP True' is the IWP in the retrieval database, 'IWP Retrieval' is the retrieved IWP, and ' $\int$  IWC' is the integral of IWC, i.e. IWP, calculated for retrieved IWC cases in a given profile. The scene corresponds to a CloudSat overpass on 22 January 2009 at approximately 04:15.

( $< \sim 50 \ \mu m$ ). Below 50  $\mu m$ , sensitivity to the true  $D_{\rm m}^{\rm IWC}$  is lost, and an a priori average is retrieved. At  $D_{\rm m}^{\rm IWC} > 600 \ \mu m$ , the negative bias amplifies and sensitivity again decreases. The behaviour seen in the 3.25 km  $D_{\rm m}^{\rm IWC}$  retrievals is similar to that seen in retrievals of column  $D_{\rm m}$  (Fig. 9 of May et al., 2024).

At an altitude of 7.25 km, presented in Fig. 8b, performance is significantly better than at 3.25 km. In particular, there is less negative bias observed, especially between 200 and 600  $\mu$ m. The overall variability of the retrievals at 7.25 km is also notably lower. Moving to 11.25 km (Fig. 8c), the upper limit of possible  $D_{\rm m}^{\rm IWC}$  values decreases to  $\sim$  600  $\mu$ m. Within the range of  $D_{\rm m}^{\rm IWC}$  spanned at 11.25 km, the retrievals display better accuracy than at the lower altitudes, and no bias is seen between 200  $\mu$ m and 350  $\mu$ m.


 $D_{\rm m}^{\rm IWC}$  retrieval performance was checked for all individual altitudes (not shown). A decrease in performance was observed for altitudes above 14 km, with a marked decrease in the correlation coefficient and an overall underestimation of  $D_{\rm m}^{\rm IWC}$  at all sizes. Across all altitudes examined, including those shown in Fig. 8, the RMSE is highest at low  $D_{\rm m}^{\rm IWC}$ . The performance plots in the upper panels of Fig. 8 indicate a decrease in performance at high  $D_{\rm m}^{\rm IWC}$ , reflected by a modest increase in RMSE.

Figure 8.  $D_{\rm m}^{\rm IWC}$  retrieval performance for altitudes of 3.25 km, 7.25 km, and 11.25 km is shown in panels (a), (b), and (c), respectively. The mean of the retrieved CDF is taken as a point estimate, and the mean, median, 16th quantile, and 84th quantile of the retrieved distribution mean are plotted as a function of the true value. The bias and the correlation coefficient r are calculated for all true  $D_{\rm m}^{\rm IWC}$  cases, taking the retrieval mean as the single estimate.

Nonetheless, RMSE does not reach the values observed for low  $D_{\rm m}^{\rm IWC}$ , because although the absolute differences increase, the relative differences are smaller.





If comparing the performance of  $D_{\rm m}^{\rm IWC}$  retrievals and IWC retrievals, it should be noted that  $D_{\rm m}^{\rm IWC}$  does not vary over orders of magnitude. Furthermore, there is limited information available on the full hydrometeor shape and size distributions, and this likely has an impact on the  $D_{\rm m}^{\rm IWC}$  retrieval performance. Still, the  $D_{\rm m}^{\rm IWC}$  retrieval exhibit the same trend — lower performance at low and high latitudes — as the IWC retrievals. The reasons are largely the same: higher signal attenuation occurring at low altitudes which reduces the number of active channels (supported by Fig. 1), a lack of signal variability leading to an inability to constrain the altitude of upper layer retrievals, and generally fewer data available. However, in the case of the  $D_{\rm m}^{\rm IWC}$  retrievals, there is a potential avenue for improvement. The 3.25 km retrievals show the lowest sensitivity and highest variability for a given true value at high  $D_{\rm m}^{\rm IWC}$ . This can be explained by the fact that, in this range, a further increase in particle size does not yield a significantly different brightness temperature. However, including some of the higher-frequency MWI channels in the retrievals could increase sensitivity to larger crystals, leading to overall improvements to the low-altitude high- $D_{\rm m}^{\rm IWC}$  cases. This sensitivity to larger crystals partially arises as a consequence of better sensitivity at lower altitudes, where higher  $D_{\rm m}$  occurs. This increased sensitivity could also translate to improvements in retrievals of high IWC at low altitudes.

An example scene of  $D_{\rm m}^{\rm IWC}$  is presented in Fig. 9. Within the range of  $D_{\rm m}^{\rm IWC}$  shown, the retrievals are similar in magnitude to the database values. The retrievals capture the cloud structures well. Although the structures retrieved in Fig. 9 appear somewhat less accurate than in the IWC cases, and there is a general underestimation at higher altitudes in Fig. 9, the latitudes spanned in Fig. 9 are much higher. Therefore, some of the performance here can be explained by generally poorer retrieval performance at high latitudes, as observed for IWP retrievals in May et al. (2024).

There is a failure to capture some areas of relatively low  $D_{\rm m}^{\rm IWC}$ , i.e.  $D_{\rm m}^{\rm IWC}$ < 100  $\mu \rm m$ . For example, the database scene has numerous small patches of low  $D_{\rm m}^{\rm IWC}$  that lie above the main cloud structures. However, these are not visible in the retrievals. This is most visible in panel (c) of Fig. 9. Cases of  $D_{\rm m}^{\rm IWC} = 0$ , which correspond to zero IWC, are retrieved as very low  $D_{\rm m}^{\rm IWC}$ . This is due to little sensitivity to such cases and therefore a tendency towards the a priori mean of  $\sim 10~\mu \rm m$ , as seen in Fig. 8.

The vertical diffusivity of cloud structures appears more pronounced for the  $D_{\rm m}^{\rm IWC}$  retrievals than for IWC. This suggests that  $D_{\rm m}^{\rm IWC}$  retrievals may have worse vertical resolution than IWC retrievals. This can also be seen in panel (c) of Fig. 9, e.g. overestimation above the cloud tops and around  $-50^{\circ}$  latitude. We again stress that IWC spans orders of magnitude and differences are amplified by extremely low IWC values, whereas  $D_{\rm m}^{\rm IWC}$  does not. Therefore, even a seemingly modest overestimation in  $D_{\rm m}^{\rm IWC}$  hints at poorer vertical resolution. The striping effect is also present here, for the same reasons as discussed for IWC, i.e. performing retrievals case-by-case. Notably, the noise appears to have an even greater impact on the retrievals than for IWC. This is evident when comparing panels (b) and (d) of Fig. 9, where the striping is particularly strong in panel (b) and entirely absent in panel (d).

As done for IWP and  $Z_{\rm m}$  in Sect. 5.2, we also compare distributions of  $D_{\rm m}$ , which can be calculated from IWC and  $D_{\rm m}^{\rm IWC}$  according to Eriksson et al. (2020). Distributions are presented in panel (d) of Fig. 6. The distribution of direct retrievals of  $D_{\rm m}$  shows good agreement to the database-derived distribution. For  $D_{\rm m}$  calculated using retrieved  $D_{\rm m}^{\rm IWC}$ , the distribution agrees in shape with the database-derived distribution, but shows larger discrepancies than for  $Z_{\rm m}$ . Agreement is relatively good in the mid-range of  $D_{\rm m}$ . For  $D_{\rm m} < 100~\mu{\rm m}$  and  $D_{\rm m} > 700~\mu{\rm m}$ , poorer agreement is seen. This negative bias is attributed to a general underestimation of  $D_{\rm m}^{\rm IWC}$  across most of its range. For example, cases of the integrated retrieved  $D_{\rm m} \sim 50~\mu{\rm m}$  were found to arise due to a strong underestimation of  $D_{\rm m}^{\rm IWC}$  at several altitudes in a profile, typically at low altitudes. These would be cases of relatively low  $D_{\rm m}^{\rm IWC}$  that lie close to or below the 16th quantile in Fig. 8, thus substantially lowering the column integral. Since  $D_{\rm m}^{\rm IWC} > 700~\mu{\rm m}$  is nearly always underestimated, the negative bias at high  $D_{\rm m}$  is likewise expected. Furthermore, unlike  $Z_{\rm m}$  which depends only on retrieved IWC,  $D_{\rm m}$  is derived from both retrieved IWC and retrieved  $D_{\rm m,IWC}$  profiles, amplifying any retrieval inaccuracies. Cases of the integrated retrieved  $D_{\rm m} \sim 50~\mu{\rm m}$  were also typically associated with low IWC, which are retrieved less accurately. Another plausible reason for the differences is that the IWC and  $D_{\rm m}^{\rm IWC}$  errors are correlated at each altitude. However, this information is not provided by QRNN.

Although the results presented in this section indicate that  $D_{\rm m}^{\rm IWC}$  can reliably be retrieved, the performance metrics are based on comparisons between database quantities and inversions of database radiances. This does not guarantee an equally low RMSE or overall accuracy as seen in Fig. 8 when applied to real ICI data. It is possible that the use of predefined PSDs in the simulations may lead to an underestimation of the true variability of  $D_{\rm m}^{\rm IWC}$ , and that the retrieval metrics reflect this. While simulations were found to be statistically consistent with observations in May et al. (2024), the extent to which real ICI retrievals will capture the full variability of  $D_{\rm m}^{\rm IWC}$  can only be evaluated after ICI's launch.

## 5.4 Vertical resolution





Some retrieval uncertainty will arise due to the physical limitation of passive sensors. Although our database contains IWC and  $D_{\rm m}^{\rm IWC}$  at 500 m levels, ICI observations do not necessarily contain enough information to resolve such fine cloud structure in

Figure 9. A comparison of  $D_{\rm m}^{\rm IWC}$  profiles, with database  $D_{\rm m}^{\rm IWC}$  shown in (a), and the corresponding retrieved  $D_{\rm m}^{\rm IWC}$  shown in (b). The retrieved  $D_{\rm m}^{\rm IWC}$  is plotted using the retrieval mean as the single estimate for each case. Panel (c) shows the difference between the retrieval and the database, relative to the database value. Only data points with either database or with retrieved IWC  $\geq 1~{\rm mg~m}^{-3}$  are shown. Panel (d) shows the retrieval performed on noise-free input data. The scene corresponds to a CloudSat overpass on 22 January 2009 at approximately 19:15.

the retrievals. A retrieval at a given 500 m level is therefore likely to be influenced by a priori information from surrounding levels.

Retrieval error correlation matrices are presented in Fig. 10. Although the retrieval errors are not strictly Gaussian, the correlation structure still captures the influence of surrounding layers on the retrievals. Both IWC (Fig. 10a) and  $D_{\rm m}^{\rm IWC}$  (Fig. 10b) exhibit a band of positive correlations that extend several layers on either side of the diagonal. Within the mid-altitude range, the error correlations for both IWC and  $D_{\rm m}^{\rm IWC}$  generally drop below  $e^{-1}$  within 2 to 2.5 km. This implies that retrievals at adjacent layers cannot be resolved fully independently, indicating that ICI's effective resolution is larger than 500 m.


While the error correlation matrices provide a qualitative insight into the relationships between vertical layers, averaging kernels offer a way of quantifying ICI's effective resolution. In this section, we compute the approximate averaging kernels associated with the retrievals. The vertical resolution of a retrieved profile can be estimated from the averaging kernel matrix  $\bf A$ , defined as the sensitivity of a retrieval  $\hat{\bf x}$  to the true state  $\bf x$ :

$$\mathbf{A} = \frac{\partial \hat{x}}{\partial x}.\tag{4}$$

Figure 10. Error correlation matrix for retrievals of IWC, in panel (a), and retrievals of  $D_{\rm m}^{\rm IWC}$ , in panel (b). The error cross-correlation matrix between IWC and  $D_{\rm m}^{\rm IWC}$  is shown in panel (c). The errors are calculated as the difference between the retrieved value and the database value. The correlation matrices are computed over all IWC and  $D_{\rm m}^{\rm IWC}$  data that correspond to a database IWP  $> 0.1~{\rm kg~m^{-2}}$ , i.e. the same threshold used for the averaging kernel calculations in Sect. 5.4. The black contour line indicates the approximate altitude below which the correlation coefficient falls below  $e^{-1}$ , or  $\sim$ 37%. Each grid cell, or matrix element, is 500 m in width and height.

Following Rodgers (2000), a retrieval estimate can be modelled as an operation on the difference between the true state and the a priori estimate with A:

$$\hat{\boldsymbol{x}} = \boldsymbol{x}_a + \mathbf{A}(\boldsymbol{x} - \boldsymbol{x}_a). \tag{5}$$

Rows of **A** are the averaging kernels which characterise the sensitivity of a retrieval at a given level to the surrounding levels. The retrieval resolution at a given level is provided by the full width at half maximum (FWHM) of the averaging kernel.

QRNN does not provide  $\mathbf{A}$  as an output. However, Rydberg et al. (2009) showed how a mean  $\mathbf{A}$  can be derived from an ensemble of cases, necessitated also by the fact that BMCI does not provide  $\mathbf{A}$  either. The average  $\mathbf{A}$  is estimated over a set of data through:

$$\mathbf{A} = \left( (\Delta \mathbf{X} \Delta \mathbf{X}^T)^{-1} \Delta \mathbf{X} \Delta \hat{\mathbf{X}}^T \right)^T, \tag{6}$$

with


$$\Delta \hat{\mathbf{X}} = (\hat{\mathbf{x}}_1 - \mathbf{x}_a, ..., \hat{\mathbf{x}}_n - \mathbf{x}_a),\tag{7}$$

and

$$\Delta \mathbf{X} = (x_1 - x_a, ..., x_n - x_a).$$
 (8)

The above theory is valid for linear retrievals, and therefore the chosen subset of data should be near-linear. Although our retrievals of IWC are not strictly linear, Fig. 9 in Grützun et al. (2018) indicates that Jacobians of IWC and snow water content (SWC) do not display strong variability and are therefore suitable for this purpose. In contrast, Fig. 8 of Grützun et al. (2018) shows that LWC and rain water content (RWC) are highly state dependent, and thus far more non-linear. However, to ensure

**Figure 11.** Averaging kernels for retrieved IWC are shown in panel (a). In panel (b), the corresponding measurement response is given as a function of altitude. The estimated retrieval resolution, estimated from the FWHM of the averaging kernels, is shown in panel (c).

near-linearity, we select only a subset of IWC data corresponding to IWP  $> 0.1~{\rm kg~m^{-2}}$  and 5 km  $< Z_{\rm m} < 12~{\rm km}$ . This case corresponds to the first row (All-cloud) in Table 1.

A log-transformation was applied to the IWC data and cases of IWC  $\leq 0.01~\mathrm{g~m^{-3}}$  were set to a random value between  $0.01~\mathrm{mg~m^{-3}}$  and  $0.01~\mathrm{g~m^{-3}}$ , with the threshold of  $0.01~\mathrm{g~m^{-3}}$  chosen to correspond to the lower sensitivity limit of the retrievals as observed in Fig. 2.  $D_{\mathrm{m}}^{\mathrm{IWC}}$  data was filtered according to the same criteria as for IWC. Cases of  $D_{\mathrm{m}}^{\mathrm{IWC}} = 0$  were replaced with  $D_{\mathrm{m}}^{\mathrm{IWC}} = 10^{-4}~\mathrm{\mu m}$ . However, it was necessary to calculate a pseudo-inverse during the computation of Eq. 6 for  $D_{\mathrm{m}}^{\mathrm{IWC}}$ , since  $(\Delta \mathbf{X} \Delta \mathbf{X}^T)$  was found to be ill-conditioned. The pseudo-inverse was calculated using a singular value decomposition (SVD), keeping only the largest elements that contribute to 90% of the variability. For both IWC and  $D_{\mathrm{m}}^{\mathrm{IWC}}$ , the analysis was also performed on different subsets of the data, i.e. varying the IWP and  $Z_{\mathrm{m}}$  criteria. The results were found to be relatively stable, achieving close to the same mean resolution.





The IWC averaging kernels for each level, the measurement response, and the estimated vertical resolution are presented in Fig. 11. The measurement response is calculated as the sum of each averaging kernel, and can be interpreted as the fraction of the retrieval that is derived from the data. A measurement response of zero implies that the retrieval is derived only from a priori information. Figure 11b shows the measurement response for IWC. Between 5 km and 15 km, the measurement response is close to, or exceeds 1.0, indicating that retrievals are derived largely from the data itself. Below the peak at 5 km, the measurement response decreases with decreasing altitude. This is due to strong signal attenuation and complex surface interactions occurring at low altitudes, causing the retrievals to be heavily influenced by a priori information. At high altitudes, the measurement response is close to zero, likely due to the low IWC present in this region, and a difficulty in constraining the altitude, as discussed in Sect. 5.1. In light of this result, the retrieval of accurate IWC estimates from real ICI observations is expected to be feasible between altitudes of 3 and 14 km.

Although Fig. 11 represents IWC retrievals at all latitudes, results were also checked for different latitudinal regions (not shown here). As expected, high latitudes produced a greater IWC measurement response at low altitudes due to the lower average altitude of clouds, and vice versa in tropical regions.

**Table 1.** Definition of the five cloud cases used for the calculation of averaging kernels.

| Case                 | Definition                                                      | IWP [g m <sup>-2</sup> ] | $Z_{ m m}$ [km] |
|----------------------|-----------------------------------------------------------------|--------------------------|-----------------|
| All-cloud            | N/A                                                             | > 100                    | [5, 12]         |
| All-cloud (extended) | N/A                                                             | > 10                     | [5, 12]         |
| Multi-layer          | $\geq 2$ layers separated by gap of IWC $< 1~{\rm mg~m^{-3}}$ . | > 10                     | [5, 12]         |
| Top-heavy            | $\frac{2}{3}$ of mass in upper $\frac{1}{3}$ of cloud.          | > 10                     | [5, 12]         |
| Bottom-heavy         | $\frac{2}{3}$ of mass in lower $\frac{1}{3}$ of cloud .         | > 10                     | [5, 12]         |

**Figure 12.** Measurement response, panel (a), and vertical resolution, panel (b), for three cloud types. The resolution is shown only for altitudes at which the measurement response exceeds 0.5. The cloud-type definitions are provided in Table 1. 'All-cloud (extended)' refers to the same cases in Table 1.

The resolution of IWC retrievals, shown in Fig. 11c, is found to vary with altitude. However, between 6 and 15 km, the resolution is relatively stable at around 2.5 km. Above 17 km, the resolution increases dramatically. ICI is not expected to be sensitive to ice clouds within this region, and thus retrievals at such altitudes are highly dependent on a priori information from the lower levels, leading to very large resolution estimates. Between the altitudes of 1 and 5 km, the resolution is smaller than 2.5 km. However, we note that the averaging kernels do not necessarily peak at the correct altitude. In other words, information is drawn from higher altitudes due to the weak sensitivity occurring at lower layers of the atmosphere. This is reflected by the low measurement response within this range. In turn, this likely contributes to the higher uncertainty observed in Fig. 2b. The low IWC resolution at low altitudes is therefore somewhat artificial, and not an indicator of success.

It is unlikely that the same vertical resolution can be achieved by all types of cloud observed by ICI. To assess the dependence of both the response and the resolution on cloud structure, averaging kernels were computed for three simple cloud classes. The cloud definitions (multi-layer, top-heavy, and bottom-heavy) are shown in Table 1. The resulting measurement response and vertical resolutions are shown in Fig. 12. The measurement response and resolution of an all-cloud case is also shown, calculated with an extended range of IWP for fairer comparison. This case is presented as 'All-cloud (extended)' in Table 1.

We note that stable calculation of averaging kernels requires a large enough subset of data. Although more cloud-types exist, further filtering quickly reduces the sample size, leading to unstable estimates. Additionally, adjusting definitions, e.g. the mass

Figure 13. Averaging kernels for retrieved  $D_{\rm m}^{\rm IWC}$  are shown in panel (a). In panel (b), the corresponding measurement response is given as a function of altitude. The estimated retrieval resolution, estimated from the FWHM of the averaging kernels, is shown in panel (c).

fractions, leads to differences in the results. Various definitions and filtering restrictions were tested and we found that overall qualitative trends can be described, but exact values can vary and should therefore not be taken as definitive estimates.





In multi-layer clouds, the measurement response resembles the all-cloud case at 5-8 km but decreases above 8 km, resulting in a bottom-heavy shape. Lower clouds with generally higher IWC increase the attenuation of higher-frequency channels that would otherwise be sensitive to the thin upper cloud. As a result, the measurement response deteriorates at higher altitudes. Vertical resolution is generally poorer, which is expected as information is drawn from neighbouring layers when an individual cloud layer is thinner than the effective resolution. Bottom-heavy clouds also show a downward shift of the measurement response shifts to lower altitudes, since the majority of the information contained in the radiances originates in the lower portion of the cloud. Resolution is, on average, slightly poorer in the lower portion of the cloud. For top-heavy clouds, response behaviour is similar to the all-cloud case, but poorer resolution was seen in the upper cloud layers, i.e. at 5-10 km. In summary, the 2.5 km effective resolution represents an average, but it is condition dependent. In cloud-specific cases, high-IWC regions produce higher responses but broader kernels, whereas low-IWC regions show the opposite behaviour.

 $D_{\rm m}^{\rm IWC}$  averaging kernels are shown in Fig. 13a. Similarly to IWC, averaging kernels outside the 5 to 15 km range did not peak at the correct altitude, but rather several km above or below that altitude. Although these averaging kernels did give reasonable measurement response and resolution results, the results can be misleading. These altitudes are therefore omitted from all panels of Fig. 13.

The measurement response, shown in Fig. 13b, exhibits the same multimodal behaviour as seen for the IWC case. However, the measurement response is lower than that of IWC at altitudes greater than 11 km. Figure 13c shows the resolution of  $D_{\rm m}^{\rm IWC}$  retrievals. The resolution at 5 km is comparable to that of IWC, achieving a resolution of 2.5 km. However, the resolution is found to degrade with increasing altitude, resulting in a resolution of over 10 km at an altitude of 15 km. This corresponds with significant decrease in measurement response at the same altitude, and the decrease in performance at altitudes over 14 km as discussed in Sect. 5.3.

# 635 5.5 Information content aspects




The degrees of freedom (DoFs) of the observations can offer further insight into the vertical information available for a retrieval. DoFs do not provide a precise measure of the resolution, nor are they accompanied by the altitude levels at which the information is derived. Nonetheless, a higher DoF value, or higher number of independent pieces of information, implies that information can be retrieved at a greater number of distinct altitudes. This connotes a better vertical resolution.

However, a higher information content can also indicate an ability to independently constrain different variables. The error cross-correlation matrix between IWC and  $D_{\rm m}^{\rm IWC}$ , shown in Fig. 10c, provides evidence that information is available to constrain the two variables separately. Although some positive correlation is observed between variables at similar altitudes, the overall correlation is weak. This suggests that total DoF of an ICI observation can likely be decomposed into contributions relating to individual variables as well as separate altitudes.

Figure 8 of May et al. (2024) presents the DoFs of ICI observations as a function of IWP and water vapour (WV). For high-IWP and high-WV conditions, i.e. deep convective clouds, ICI observations achieve a DoF of between 8 and 10. The atmospheric column of interest can be assumed to be around 10 km thick, i.e. a retrieval domain between 5 and 15 km. The total DoFs represent information available for all variables. This includes IWC,  $D_{\rm m}^{\rm IWC}$ , and — to a lesser extent — humidity, LWC, and rain. Taking these factors into account, the maximum DoF suggests a potential vertical resolution of about 2 to 3 km for IWC and  $D_{\rm m}^{\rm IWC}$ , which supports the results presented in Sect. 5.4.

Grützun et al. (2018) calculates the reduction of degrees of freedom ( $\Delta DOF$ ) as a quantification of the number of pieces of information obtained from an ICI measurement, where the maximum possible  $\Delta DOF$  is 11, i.e the number of ICI channels, neglecting polarisation. The mean total information content for ICI is estimated at 6.19. When the information content is decomposed into contributions from individual variables,  $\Delta DOF$  for IWC was estimated to be approximately 4. Based on this, Grützun et al. (2018) speculated that it may be possible to estimate IWC profiles containing independent information at four distinct altitudes. It is important to note that Grützun et al. (2018) defines IWC as the cloud ice mass density, whereas the density of frozen precipitation is defined separately as SWC. In contrast, our definition includes all frozen hydrometeors—both in-cloud and precipitating. Considering only altitudes between 5 and 15 km, the four distinct levels hypothesised by Grützun et al. (2018) imply a vertical resolution of  $\sim$ 2.5 km.

To represent the microphysical properties of cloud ice hydrometeors, Grützun et al. (2018) used the particle mean mass. An information content of 2.70 was estimated for mean mass, which is again consistent with our slightly higher resolution estimate for  $D_{\rm m}^{\rm IWC}$ . They also found that simultaneously measuring mean mass and mass density reduces the information content of the mean mass retrieval, due to correlations between the two variables. This finding—that microphysical characteristics can be derived using information partly independent from that used for IWC, albeit with some loss in the total information content when both are retrieved together —supports the low cross-correlation found in Fig. 10c.

Similar findings are reported by Pfreundschuh et al. (2020), where DoFs are calculated from the trace of the averaging kernel matrix. In the example cases presented, the total DoF did not exceed 10, despite the use of both MWI and ICI observations in the measurement vector. Decomposing the DoF into contributions from particle size and particle concentration dimensions showed

that different retrieval quantities contribute separately to the total information content. Together, the results from Grützun et al. (2018) and Pfreundschuh et al. (2020) suggest that the vertical resolution of our  $D_{\rm m}^{\rm IWC}$  profile retrievals arise from some independent information, rather than just constraints from the IWC retrieval.

We can therefore conclude that, by virtue of the multiple frequencies at which ICI measures, there can be enough information in an observation to constrain the retrievals of both IWC and  $D_{\rm m}^{\rm IWC}$  to some extent independently. As a result, ICI may have capabilities going beyond single-frequency cloud radars. Despite the fact that a radar has a high number of DoFs relative to passive instruments, such as ICI, each independent piece of information corresponds to a single altitude level. Therefore, since radar's high DoFs are only associated with high vertical resolution, this information content does not necessarily translate into an ability to retrieve multiple variables at a given altitude. In fact, decomposing the DoFs of radar into contributions from individual retrieval quantities shows that a single variable dominates (Pfreundschuh et al., 2020). Consequently, a priori assumptions must play a significant role if both IWC and  $D_{\rm m}^{\rm IWC}$  are to be retrieved from radar measurements. If retrievals of ice mass are derived from both radar and lidar observations, as is the case for DARDAR and 2C-ice, then the total available information across an entire column may increase. However, lidar is not available for the entire column, particularly in the case of high IWC. Therefore, at a given altitude, the number of independent pieces of information will not necessarily be higher.

## 5.6 Sensitivity analysis





The uncertainty in the retrievals could arise from three sources. Firstly, there are physical limitations which may affect the sensitivity of the observations to the atmospheric state. Secondly, there are limitations in our knowledge of the physical system, which take the form of assumptions made in the radiative transfer calculations. Finally, there will always be inherent uncertainty due to noise. It is difficult to identify exactly to what extent the uncertainties arise from each of these sources. However, one approach is to retrain the retrieval model under different conditions and compare the retrieval performance.

The primary physical limitation is the true vertical resolution. Since the true resolution is likely closer to 2.5 km than to 500 m, as shown in Sect. 5.4, we hypothesised that some retrieval inaccuracy arises from the model's difficulty in distinguishing between neighbouring layers, and retrieving on a coarser altitude grid may help to avoid these issues. To examine this possibility, IWC profiles in the database were averaged over 2 km layers. The model was then retrained to retrieve IWC profiles at a 2 km resolution.

A comparison of retrieval performance between the two models — original 500 m resolution and new 2 km resolution — is presented in Fig. 14a. Very little difference can be seen between the two models, aside from a small improvement of the mean and lower quantile at low IWC. Improvements as a result of using a lower resolution model may be expected in cases of multiple layers of clouds separated by a thin region of little to no ice. If this layer is thinner than the effective resolution of the observations, the 500 m model would not be able to resolve this layer. Instead, information from the surrounding cloud layers would leak into the retrieval at this altitude and overestimation would occur. Upon checking individual retrieved profiles for both the 500 m and 2 km models (not shown), this was confirmed to occur. An example is presented in Fig. 15b. However, in the case of a single cloud layer, with few small details to resolve, both models will largely behave the same, e.g. in Fig.

15a. Since cases with a single cloud layer were found to be far more common in the database, improvements to layered-cloud profiles have a minimal effect on the overall performance statistics, explaining the similar performance of both models.

At low-to-moderate IWC, the 2 km model shows reduced underestimation. Upon investigation of individual profiles, this appears to occur when the true IWC profile contains a sharp but narrow peak. An example is shown in Fig. 15c. The 500 m resolution retrieval model fails to retrieve this feature. However, averaging over 2 km layers smooths out the peak, lowering the reference IWC. Therefore, even if the two retrieval models predict the same IWC at a given altitude, the retrieval error of the 2 km model is smaller due to the lower reference IWC. In this case, the improvement in the lower quantile achieved by the 2 km model does not in fact indicate better model accuracy relative to the truth.







A risk of using a high-resolution model when the effective resolution of the observations is lower is overfitting, i.e. the model 'guesses' small IWC details that are not present in reality. Since the 500 m model does not overestimate IWC compared to the 2 km model, this does not appear to be the case. Therefore, the main conclusion of this sensitivity analysis is that, despite some small differences, the two models achieve very similar performance across the range of IWC. Although retrieving at a coarser resolution is more in-line with the physical limitations of the observations, there seems to be no benefit to retrieving at a courser resolution, since using a finer grid of 500 m does not harm performance. Any small advantages shown by the 2 km model are not common enough to have a significant impact on the overall statistics, and do not influence the mean or median in any one direction. This suggests that the true resolution, estimated at around  $\sim 2.5$  km, is already inherently captured by the retrieval model.

Furthermore, the error correlations in Fig. 10a offer further explanation as to the similar performance seen for both the  $500 \mathrm{\ m}$  and  $2 \mathrm{\ km}$  resolution models. Since errors are seen to be correlated over distances more than  $2 \mathrm{\ km}$ , retrievals at a  $500 \mathrm{\ m}$  resolution are not giving totally independent information, and therefore the precision is not improved by retrieving at  $2 \mathrm{\ km}$ . As a result, the two models likely have similar vertical smoothing and therefore comparable retrieval accuracy.

The choice of particle model used in a simulation may also impact the retrieval of IWC from the given simulation. The sensitivity of IWP retrievals to the particle model was investigated in May et al. (2024), where it was suggested that the retrieval model could somewhat distinguish between particle model, but assumed some mean level of extinction that translated to over- or under-estimation of IWP. Drawing from the conclusions made in May et al. (2024), it is therefore reasonable to expect that some inaccuracy in the IWC retrievals may arise if the retrieval model assumes a mean particle model.

To explore this possibility, a model (denoted 'AA1 model') was trained with only simulations generated using the AA1 particle model. This corresponds to 30% of the cases used to train the original model. At low IWC, the performance of the AA1 model is similar to the original model, as shown in Fig. 14b. There are small differences, but these could be attributed to the use of a smaller dataset to train the AA1 model and a smaller dataset on which the inversions were performed.

At higher IWC, the AA1 model shows better performance compared to the original model, i.e. less variability for a given true value and lower MFE. The mean and median shift closer to the identity line, and the quantiles become more symmetric around the line. The particle models 'Snow' and 'AA2', defined in see Table 2 of May et al. (2024), are likely to cause an underestimation of IWC when the actual model is AA1. A further discussion of this possibility is given in Sect. 5.5 of May

**Figure 14.** IWC retrieval performance for all altitudes for alternative model runs. In panel (a), the original model is compared against a model trained on a version of the database with vertical resolution reduced to 2 km, denoted 'New (2km)'. In panel (b), the original model is compared to a model trained (and retrieved) only on cases simulated using the 'AA1' particle model, denoted 'AA1 model'. For comparison, the original model was also used to retrieve only cases simulated with the 'AA1' particle model, and the performance is denoted in the plot as 'Original model (AA1)'.

et al. (2024). The improved performance of the AA1 model therefore suggests that the increased spread in retrievals for a given IWC, seen for the original model, partially stems from the model's difficulty in reliably distinguishing between particle models.

#### 5.7 Limitations


The retrieval results presented in this article are performed and compared to a subset of the retrieval database. Since the database relies on information from CloudSat and MODIS, it is therefore important to consider how the limitations of this data may impact future retrievals on real ICI data. Specifically, whether the sensitivity range of these instruments may lead to a difficulty in detecting certain cloud types.

For instance, CloudSat-based retrieval products also typically incorporate CALIPSO lidar data in order to better characterise high clouds, such as thin cirrus. Since lidar information is not used for the ICI retrieval database, many thin cirrus cases may therefore be absent in the database and, by extension, the retrievals. However, even if such cases were included in the database, they would likely not be retrieved reliably. We find that ICI's lower sensitivity limit to IWC is around 10 mg m<sup>-3</sup>. In contrast, radar-only based retrievals in DARDAR include cases down to 1 mg m<sup>-3</sup> (Cazenave et al., 2019). In future iterations of the database, EarthCARE is expected to replace CloudSat. Although EarthCARE will offer higher sensitivity than CloudSat, the lower sensitivity limit of ICI will likely still lead to an underestimation of the number of thin cirrus cases if used operationally.

**Figure 15.** Example IWC profiles for the original 500 m resolution retrieval model and the 2 km resolution retrieval model. Dashed lines represent the truth, i.e. IWC profiles from the ICI retrieval database, where the 2 km resolution profile is simply an average of the original profile over 2 km layers.

If considering retrievals of IWP, DARDAR tends to saturate at an upper limit of 10 kg m<sup>-3</sup>, whereas our retrieved IWP CDFs extend up to ~ 40 kg m<sup>-3</sup>. Similarly, 2C-ICE has also been shown to achieve higher IWP on average (Pfreundschuh et al., 2025). Extending to retrievals of IWC, the same trend between ICI and DARDAR is demonstrated in Fig. 4. Therefore, it does not appear that ICI is limited by the CloudSat data in terms of saturation in high-IWC regions. However, it must be acknowledged that multiple scattering in high-IWC regions will decrease the radar signal. In turn, this leads to fewer high-IWP cases in the database. Although ICI observations may be able to detect higher IWC than radar, and higher values are certainly seen in comparison to DARDAR, fewer high-IWC cases will be available for the retrieval model to train on.

There are also caveats to a machine learning approach. A key challenge is understanding exactly how the model derives its predictions. In contrast, methods such as OEM explicitly provide the relationship between observations and the state variables, e.g. through the Jacobian. These relationships must instead be approximated if neural networks are implemented, as described in Sect. 5.4. Additionally, neural networks cannot represent correlations between multiple outputs that exist in reality, such as those between layers of IWC. Developing an approach to sample from retrieved distributions such that correlations are captured is therefore a potential avenue for future research.

# 6 Summary and conclusions


The motivation behind this study was to explore ICI's potential as a source of vertical ice mass information. The study focuses on two key questions: How well can ice mass profiles be derived from ICI observations? Can ICI act as complementary to existing products, such as those derived from radar and lidar measurements?

In this study, we retrieved IWC and profiles of mean mass diameter  $D_{\rm m}^{\rm IWC}$  from ICI observations. The retrievals were performed using a QRNN trained on simulated ICI observations. These same simulations also constitute the ICI retrieval database to be used at EUMETSAT for operational ICI retrievals. As such, they represent the state-of-the-art in radiative transfer calculations at microwave and sub-millimetre wavelengths. By using the same data intended for operational L2 column variable retrievals, we demonstrate that an ICI L2 product containing information on the vertical ice mass information is also achievable.

The results of this study show that IWC and profiles of mean mass diameter  $D_{\rm m}^{\rm IWC}$  can be reliably retrieved from ICI observations. IWC retrievals were shown to be reliable within the range of 0.01 and 1 g m<sup>-3</sup>.  $D_{\rm m}^{\rm IWC}$  retrievals performed well within the range of 25 to 600  $\mu$ m. Retrievals at altitudes of 3.25 km, 7.25 km and 11.25 km are presented in this paper, with the highest performance observed at 11.25 km. For both IWC and  $D_{\rm m}^{\rm IWC}$ , retrieval performance was observed to vary with altitude, attributed primarily to variation in signal attenuation across altitudes. Outside of the range of 3 to 14 km, retrievals are deemed to be inaccurate due to reduced sensitivity of the observations to ice mass at these more extreme altitudes. Some instability was observed between neighbouring profiles, arising as an artefact of retrieving each profile individually. This effect was removed when performing the same retrievals on noise-free radiances, revealing that instrument noise has a discernible effect on retrievals, particularly for low IWC.





A comparison of our IWC retrievals to the DARDAR product displayed statistical consistency. Furthermore, integration of the IWC retrievals produced IWP that is statistically consistent both with DARDAR and with IWP as expected from the ICI L2 product. These findings suggest that the IWC retrievals presented in this study are reliable and, in the future, could serve as a complementary data source to EarthCARE. Furthermore, ICI could prove even more effective than a radar at retrieving  $D_{\rm m}^{\rm IWC}$  due to ICI's broader frequency range, as discussed in Sect. 5.5.  $Z_{\rm m}$ , derived using retrieved IWC, showed good agreement with direct retrievals of  $Z_{\rm m}$ . Larger differences were observed when comparing direct retrievals of  $D_{\rm m}$  with  $D_{\rm m}$  derived from IWC and  $D_{\rm m,IWC}$ , highlighting a need to still retrieve column values in parallel to profiles.

The retrievals were performed at a resolution of 500 m, though the effective resolution of the retrievals is believed to be poorer. Averaging kernels were derived for the first time for this type of observations. The methodology of Rydberg et al. (2009) was applied, providing only mean averaging kernels for relatively large ensembles of retrievals performed on simulated data. Since averaging kernels are a highly useful tool for characterising retrievals, improved approaches for deriving this information would be beneficial. Averaging kernels were approximated over a subset of the data, allowing the effective resolution of the IWC retrievals to be estimated at around 2.5 km. The effective resolution of  $D_{\rm m}^{\rm IWC}$  retrievals is comparable to IWC at low altitudes (a resolution of 2.5 km at an altitude of 5 km), but resolution was found to become poorer with altitude. The retrieval error correlation matrices also suggest an effective vertical resolution of around 2 to 2.5 km, which is consistent with our estimations derived from averaging kernels. The IWC averaging kernel analysis was applied to several basic cloud-types, where higher-IWC regions of cloud were found to yield a higher response function but broaden the kernels. Multi-layer clouds achieved a slightly poorer resolution than 2.5 km. However, retrieving IWC at a resolution of 500 m was not found to weaken the performance of the retrievals in a significant sense. In contrast, microphysical assumptions in the database simulations were

identified as contributing to a reduction of performance, although their contribution was found to be more pronounced at the upper end of ICI's sensitivity range to IWC.

A significant step in this research will be the launch of ICI, allowing for the retrieval of profiles from real ICI data. In this study, we aim to identify potential sources of uncertainty within the retrievals, but the use of real observations may reveal additional limitations. In particular, it allows us to better validate our database simulations, which affect the quality of our retrievals. For example, we will be able to assess the success of the surface emissivity scheme used in the database generation scheme, presented in May et al. (2024). In turn, this comparison may allow us to better understand the uncertainties associated with low altitude IWC and  $D_{\rm m}^{\rm IWC}$  retrievals. Excitingly, with the launch of ICI also comes the possibility to validate our retrievals with existing products. While ICI will not be colocated with any radar/lidar measurements, it will be possible to compare to machine learning-based retrieval products using geostationary sensors (Amell et al., 2024). Our use of a probabilistic machine learning model also provides a method for quantifying retrieval uncertainties. This is particularly valuable to applications such as climate model verification, especially in light of the fact that ICI will offer observations spanning 22 years. Finally, the launch of ICI also signals the launch of MWI. There will therefore also be the possibility to incorporate data from MWI into ICI retrievals. This will be particularly beneficial for retrievals at low altitudes, where the largest  $D_{\rm m}$  can be found.

The launch of AWS (Eriksson et al., 2025) marked an important milestone, providing the first operational sub-millimetre observations of atmospheric ice. Preliminary comparisons of AWS- and ICI-based retrievals of column variables indicate that AWS performs comparably well for IWP (not shown). Since it is possible to retrieve ice mass profiles from ICI's measurements, it can likely also be achieved with AWS. However, AWS shows weaker retrieval performance for  $Z_{\rm m}$  and  $D_{\rm m}$ . This can be attributed to the fact that AWS lacks channels at 448 and 664 GHz. Therefore, it is unlikely that the same quality of profile retrievals would be achieved. Nonetheless, AWS offers many of the same benefits we anticipate from ICI in the assessment and validation of our simulation framework, including testing of profile retrievals on real satellite data. Given that AWS is already launched, these opportunities are possible immediately. Both ICI and AWS offer exciting potential for ice mass retrievals, contributing to a better understanding of atmospheric ice in Earth's climate system.

Code and data availability. The code used for analysis and plotting is available at https://zenodo.org/records/15374048 (May, 2025). The data pertaining to ICI used in this study are available under license for non-commercial purposes and on the condition of no redistribution by contacting EUMETSAT (vinia.mattioli@eumetsat.int). DARDAR-cloud v3-10 data are available at https://www.icare.univ-lille.fr/dardar/data-access/ (last access: 19 March 2024).

## Appendix A: Neural network architecture





The QRNN implemented in this study takes the following inputs:

– Antenna-weighted brightness temperatures  $T_a$  for each of the 13 ICI channels. Measurement noise is generated by randomly sampling from a zero-mean Gaussian distribution with variance of 75% of the NE $\Delta$ T estimates for ICI Eriksson

et al. (2020). New noise values are sampled for each input  $T_a$  in each batch of data and each epoch during training. This prevents the model from encountering the same noisy values more than once during training.

- Surface type classification, surface temperature, and surface pressure are included as ancillary data.

The QRNN outputs predictions for 83 independent variables. These variables consist of column integrated variables — IWP,  $D_{\rm m}$ , and  $Z_{\rm m}$  — along with 40 levels of IWC and 40 levels of  $D_{\rm m}^{\rm IWC}$ , both at a 500 m resolution. For each of the 83 output variables, the model is trained to predict 17 uniformly spaced quantile levels  $\tau \in [0.01, 0.99]$ .

A log-linear transformation was applied to the IWP and IWC data to account for the wide range of magnitudes spanned by these variables. IWP and IWC values less than 1.0  $\rm ~kg~m^{-2}$  were transformed into logarithmic space. Cases of IWP  $\rm < 10^{-4}~kg~m^{-2}$  were replaced with a random sample between  $\rm 10^{-6}~kg~m^{-2}$  and  $\rm 10^{-4}~kg~m^{-2}$ . Cases of IWC  $\rm < 10^{-6}~kg~m^{-3}$  were replaced with a random sample between  $\rm 10^{-9}~kg~m^{-3}$  and  $\rm 10^{-6}~kg~m^{-3}$ . The remainder of the output data were linearly normalised according to the range of IWP covered in the training set.

*Author contributions.* EM led the writing of this paper, conducted the retrievals, and performed the visualisation and analysis of the results. PE provided scientific input on all results and contributed to revisions of the manuscript.

Competing interests. The contact author has declared that none of the authors has any competing interests.


Acknowledgements. The ICI simulations that enabled this study were performed as part of a EUMETSAT study "Development of a cloud radiation database for EPS-SG ICI". The authors would like to thank Simon Pfreundschuh for the development of the QRNN Python package, available at https://github.com/simonpf/quantnn (last access: 31 January 2025), which was used for the retrievals in this study. The retrievals were performed using computational resources at the Chalmers Centre for Computational Science and Engineering (C3SE), provided by the Swedish National Infrastructure for Computing (SNIC).

Financial support. The research was mainly funded by the Swedish National Space Agency (grant no. 2021-00077). Development of the ICI retrieval database was performed under the EUMETSAT study "Development of a cloud radiation database for EPS-SG ICI" (contract EUM/CO/21/4600002601/VM).

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
