# Peer review of "The Ice Cloud Imager: retrieval of frozen water mass profiles"

_EGUsphere, 2025_

## Author Comment (AC1)

**Response to Referee #1**

September 26, 2025

We thank the reviewer for their thoughtful feedback and for raising several important topics. Each point is addressed below.

**Specific comments**

In the Introduction, the authors provide the rationale for such a method and for retrieving the vertical profile of ice water content instead of estimating the ice water path in the column. One of the arguments they present is the radiative effects of cloud ice, and I agree with this in general. It is clear that the same mass of ice can be distributed within the cloud limits in a number of ways, and the radiative transfer and radiative effects for these distributions will not be the same. For example, the emission depends on temperature, and a cloud with top-to-bottom IWC(z) falloff will not be equivalent to a bottom-to-top IWC(z) falloff, despite the fact that their IWPs are the same. However, it has already been shown using the same DARDAR dataset and DISORT calculations that the absolute differences for short-wave and long-wave fluxes estimated with and without knowledge of IWC(z) shape do not exceed 2 W/m2 at the top of the atmosphere, 2.7 W/m2 at the surface, and 4 W/m2 in the atmosphere. If these results are cloud amount weighted, these values reduce to 0.5 W/m2, 0.5 W/m2, and 1 W/m2, respectively. From this point of view, it would be useful to provide an example of a real physical situation for which the error of using constant IWC instead of a real IWC(z) profile would lead to misinterpretation of a physical phenomenon or model validation.

- The results mentioned are new to us, and so we thank the reviewer for bringing them to our attention; they are certainly interesting and relevant to our study.

  However, it is not clear to us from the results cited whether the small net flux differences are independent of the cloud height, or whether they were obtained only by comparing to a constant IWC for clouds placed at the same height. We assume that the results would vary depending on cloud-top and cloud-base heights, since the longwave emission will depend on temperature. If this is the case, this information is still important. Unlike passive thermal infrared measurements, ICI's radiances are not directly related to the physical cloud top temperature, and must be retrieved. Therefore, retrievals of IWC remain important because they provide estimates of the cloud-top and -base heights, which cannot be inferred otherwise.

  Additionally, knowledge of the IWC profile is needed to constrain radiative heating rates inside the cloud. Hartmann and Berry (2017) shows that TOA radiation balance is similar for two anvil clouds with the same IWP but different shapes, which is consistent with the findings that the reviewer reports. However, heating rates displayed distinctly different behaviour for the two cloud types. Likewise, ? finds that while surface and TOA fluxes are relatively insensitive to changes in IWC or particle size, the averaging heating rate can vary by 10-20% between 8 and 14 km. Radiative heating is a driver of vertical motion, and impacts convective cloud systems and large-scale circulation. As such, knowledge of the real IWC($z$) profile is needed for its accurate calculation.

- **Changes to the manuscript:** The following sentences have been added to the manuscript: "The vertical distribution of ice has also been shown to impact radiative heating rates (Mather et al., 2007; Hartmann and Berry, 2017), which drive convective systems and large-scale circulation.", "Unlike passive thermal infrared measurements, microwave radiances are not directly related to the physical cloud top temperature, which is needed for long-wave flux estimates, and must be estimated from retrieved vertical information."

Section 3.2. Retrieval model implementation

I am not an expert in neural network training, but my experience with forward and inverse problems tells me that adding noise to the simulated radiance increases the chances of retrieving an incorrect original profile, especially in the case of an ill-posed problem. Indeed, this is a typical self-consistency test in any method, for which the input data are passed through the forward simulator, then modified by realistic noise, and then passed through the retrieval procedure to compare with the reference data. However, I'm not sure that the training dataset should be modified by noise. It's true that the real data will be noisy, but the training process using noise-free data should yield similar weights to the neural network's nodes and paths as a noise-perturbed one, but this training will take less time/data and be more physical. Later on, its accuracy will be reduced by using noisy data, but the neural network itself will be "cleaner". Could you please comment on this? Will one still require 9.4 million cases to train the neural network (line 216), or can one achieve the same results using 10 times fewer profiles, but without noise?

- To clarify: we did not add noise to the simulations beforehand, but rather generated a new noise value each time a training sample passed through the network. In this way, although the same training sample is seen by the model multiple times during training, each time there is a slight variation, and therefore this acts as data augmentation. Seeing slightly different variants of the same base sample during training helps the model to generalise better to unseen, noisy inputs.

  It may sound counter-intuitive to train on noisy inputs, but our choices are motivated by numerous studies showing that repeatedly adding new noise *during* training improves the robustness of the model and prevents overfitting (Zur et al., 2009; Piotrowski and Napiorkowski, 2013).

  If it were the case that, as you suggest, fewer cases would be needed to train the model in the absence of noise, this does not replace the need to cover the variability of the state space. The requirement for millions of cases is also motivated by the need to cover all possible conditions, e.g. very high-IWC cases or specific surface types. For this reason, we would still expect an equally high number of simulations to be necessary in order to fully capture real-world variability.

- **Changes to the manuscript:** To clarify our approach, we have added a short explanation in Section 3.2 describing how noise is added during training, and motivating its use as a form of data augmentation.

Lines 199-200: Indeed, the retrieval of Dm and Zm does not make sense if ice water path equals zero, but this is somewhat evident. Could you please rephrase these sentences?

- **Changes to the manuscript:** The sentence has been updated.

5.1. Retrieval Ice Water Content

In this section, the authors spend considerable time explaining the effects related to averaging kernels, but they do not mention them explicitly, despite the fact that they show them in Fig. 12 and Fig. 13. I would say that the text of this section could be made much more compact and understandable for the reader if the authors moved these figures here.

- We thank the reviewer for this suggestion. Our intention with Section 5.1 is to present a broad overview of retrieval performance that is accessible to a range of readers, while Section 5.4 acts as a more in-depth exploration on why we see the results presented earlier. While it is true that Figs. 12 and 13 would help to explain some of the results in Section 5.1, moving these figures earlier would pre-empt the derivations, motivations, and assumptions made in Section 5.4, and may risk confusing readers who are less familiar with averaging kernels (particularly since these are *approximations* of averaging kernels). Alternatively, moving the entire section earlier would risk breaking the logical progression from *what* the retrieval achieves to *why* it behaves that way. That said, we agree that it may help some readers to see the connection sooner.

- **Changes to the manuscript:** In several places in Section 5.1 we now note that some retrieval behaviours are consistent with the averaging kernel results, and refer the reader to Section 5.4.

Fig. 3, 7, 9, 10: It would be interesting to see the differences between the reference and test panels either in absolute or relative values in a third (added) panel. I am somewhat concerned about the striping mentioned in this section. Wouldn't it be better to smooth/denoise the input data to avoid this effect? Perhaps one could run the retrieval twice – once for original profiles and once for smoother ones – and if the results differ strongly, then use the second solution.

- Smoothing the input data is unlikely to have an effect on the striping artefacts, since they arise from observation noise rather than structures in the input data. Because our retrievals treat neighbouring profiles independently, and because the measurement noise is uncorrelated between observations, adjacent retrievals can fluctuate slightly up or down, producing the visual striping. As evidence for this, we have added a new panel to Figs. 3 and 9 to show the same retrievals, but performed using noise-free radiances as input. The striping disappears.

  Although the striping is only a noise artefact, we agree that it is undesirable. We thank the reviewer for their suggestion of smoothing, since this could still be an effective approach post-retrieval. However, the trade-off—-whether smoothing were to be applied pre-training or post-retrieval—-is a deterioration of the horizontal resolution. Since the horizontal resolution of ICI is not particularly high to begin with, smoothing would lead to a relatively poor resolution. Alternatively, a future avenue for this study would be to perform more advanced retrievals of the entire swath, which would likely also remove the striping.

  Our aim in this study is to present the retrieval capabilities of ICI and to identify where it succeeds and where it struggles. As such, we prefer to present unsmoothed retrievals, rather than 'hide' its occurrence from the reader. If an operational product were to be produced, we believe that it would be best left to the users to smooth the data, since this would be straightforward to do. In this case, the users can decide which has a higher priority based on the use case — lower noise/less striping or higher spatial resolution.

- **Changes to the manuscript:** A panel has been added to Figs. 3, 7, and 9, showing the difference between the retrieval and the truth, relative to the truth. The discussion has been updated to briefly discuss these panels.

  Additionally, Figs. 3, 7, and 9 have a new panel showing the same retrievals but performed on noise-free input data, in order to more clearly show the impact of the noise on the retrievals and thus explain the striping effect.

  Multiple new panels have now been added to three of the figures, due to both this suggestion and the suggestions of the other reviewer. To avoid the manuscript becoming too lengthy as a result, we have decided to remove Fig. 10 from the manuscript. Fig. 10 did not add any new information that Fig. 9 did not already provide.

**References**

Hartmann, D. L. and Berry, S. E.: The balanced radiative effect of tropical anvil clouds, Journal of Geophysical Research: Atmospheres, 122, 5003–5020, https://doi.org/10.1002/2017JD026460, 2017.

Mather, J. H., McFarlane, S. A., Miller, M. A., and Johnson, K. L.: Cloud properties and associated radiative heating rates in the tropical western Pacific, Journal of Geophysical Research: Atmospheres, 112, https://doi.org/10.1029/2006JD007555, 2007.

Piotrowski, A. P. and Napiorkowski, J. J.: A comparison of methods to avoid overfitting in neural networks training in the case of catchment runoff modelling, Journal of Hydrology, 476, 97–111, https://doi.org/10.1016/j.jhydrol.2012.10.019, 2013.

Zur, R. M., Jiang, Y., Pesce, L. L., and Drukker, K.: Noise injection for training artificial neural networks: A comparison with weight decay and early stopping, Medical Physics, 36, 4810–4818, https://doi.org/10.1118/1.3213517, 2009.

---

## Author Comment (AC2)

**Response to Referee #2**

September 26, 2025

We thank the reviewer for their careful reading and constructive suggestions. We address each point below.

**Major concerns**

Since your ML prediction actually predicts the distribution (or quantiles), it is a pity that the discussion on how to use the quantiles to develop a IWC or Dme flagging algorithm, especially given the fact that your results contain so many small IWC/Dme values that are apparently unreal but just ML artifacts because the training focuses on learning the distribution. For example, the fake "near-empty" clouds near the freezing layer in Fig. 3 case, the much larger integrated IWC value compared to your retrieved IWP in the clear-sky regime in the Fig. 7 case, the spike in Fig. 6, and the "better-than-CloudSat" in the lower sensitivity threshold suggested in your Fig. 4 PDF comparisons. My guess is the PDF width from your prediction for these small IWC cases should be larger than your retrieved IWC value, but as the errorbar was never used for filtering, I don't know if that's the case or not. Ultimately if these become operational or research products for ICI, you'll be required to provide a quality flag or something similar to let the user know which retrievals are not trustworthy. My suggestion is to try playing with different thresholds (e.g., standard deviation, 75th quantile – 25th quantile, etc.) to develop a flagging algorithm and show the confusion matrix to demonstrate both clear-sky and cloudy-sky are accurately captured. Also, please use the flagging mechanism to update Fig. 3, 4, 5, 6, and 7.

- We thank the reviewer for this suggestion, which highlights a strength of our retrieval approach — the retrieval of quantiles of the PDF. We tested at thresholds based on quantile spreads, relative to the distribution mean. The 75th-25th spread was too narrow to flag problematic cases reliably, but the 99th-1st spread showed that the low-level clouds have the highest uncertainties, indicating that their predicted distributions are strongly skewed. In this scene, rain was present underneath the low-level clouds, potentially explaining their occurrence.

  We agree that such measures could form the basis of a flagging algorithm. However, our primary aim of this study is to demonstrate the capabilities and limitations of ICI profile retrievals, rather than develop an operational product. For this reason, we prefer not to apply filtering to the main results, as this would risk 'hiding' problematic cases from the reader. However, the findings resulting from this suggestion remain valuable, and should be included.

  We also note that excluding cases would affect the overall statistics of the retrievals. For this reason, we believe that the decision of whether and how to filter should be left to an end-user, depending on the application.

- **Changes to the manuscript:** Fig. 3 has been updated to include an additional subplot showing the spread between the 1st and 99th quantile, relative to the mean. The discussion around Fig. 3 has been extended to discuss the high uncertainties associated with the low-level clouds, and to describe how this metric could be used for a flagging algorithm in future applications.

With the same IWP value, the clouds could be top-heavy (i.e., developing), U-shape (i.e., mature), or bottom-heavy (i.e., decaying). The scientific value of profile retrieval mainly lies in being able to differentiate cloud vertical structure, and potentially understand better the system life stage. The three cases shown in Fig. 15 demonstrate that your algorithm could achieve this capability. However, the averaging kernel and DoF discussions all focusing on the mean vertical resolution for all training samples. I would strongly recommend updating the averaging kernel results for different types of cloud. Given the fact that your training samples are big, you can use some clustering method (e.g., PCA, k-clustering) to separate them into a few representative types, and then compute the results for each cloud type. There is no way that the 2.5 km vertical resolution can be achieved for all kinds of ice clouds between 5-15 km, so it would be much more appreciated if readers can be informed the real physical resolution that can be achieved for different cloud types. I'm especially interested to see how multi-layer clouds can be resolved in your profile retrievals.

- It is indeed true that different resolutions will be achieved with different cloud types. We explored the three suggested cloud types (multi-layer, top-heavy, and bottom-heavy) using simple physical rules (given in Table 1 of the manuscript). We considered using a clustering method, but mapping the clusters cleanly onto physically-interpretable cloud-types could be difficult. The averaging kernels were computed for each case and presented in the manuscript.

  Since there is no unique definition for each cloud-type, adjusting, for example, the mass fractions led to differences in the results. Also, IWP, IWC, and $Z_{\mathrm{m}}$ are already restricted to ensure near-linearity, so further filtering leaves few cases, causing instability in the results. Therefore, exact values were unstable, but overall qualitative trends can be described. Results are presented in the manuscript.

  Ideally, we would apply further restrictions, e.g. looking at only top-heavy clouds with IWP > 1 kg m$^{-2}$ to target anvil clouds. However, not enough cases remain to produce stable results.

- **Changes to the manuscript:** A new figure has been added to the manuscript (Fig. 12), showing averaging kernels derived for the three cloud classes suggested by the reviewer. The discussion has been extended to present these new results. A new table has also been added (Table 1) to clearly present the five subsets of IWC now used in the averaging kernel calculations. The conclusion now includes the new results. Regarding averaging kernels, we also made very minor changes to the original IWC averaging kernels, including labels and a small change in some resolutions due to a code error. The conclusions remain the same.

**Minor points**

As mentioned in this work, the operational ICI products include mean mass height Zm and mass-weighted column averaged Dme. Could you check if your retrieved IWC can give you the mean mass height that's consistent with Zm, and your Dme and IWC profile retrievals can yield agreement with mass-weighted column averaged Dme? I'm especially curious about the former.

- To check this, we produced distributions of $Z_{\mathrm{m}}$ derived in three ways: from database IWC, from retrieved IWC, and a direct retrieval of $Z_{\mathrm{m}}$. Similar distributions were made for $D_{\mathrm{m}}$, deriving the values from both IWC and $D_{\mathrm{m, IWC}}$ profiles. The results show better agreement for $Z_{\mathrm{m}}$ than for $D_{\mathrm{m}}$. However, this is somewhat expected since $D_{\mathrm{m}}$ must be calculated using both retrieved IWC and retrieved $D_{\mathrm{m}}$ profiles, which can amplify existing inaccuracies.

- **Changes to the manuscript:** Fig. 6 has been extended to include two extra panels displaying distributions of $Z_{\mathrm{m}}$ and $D_{\mathrm{m}}$. The text has also been updated to discuss the results seen in the updated plots.

Your ML retrieval results suggest degradation happens above 12 km, but later on your averaging kernel experiments find the vertical resolution can be achieved stably below 15 km. Why this discrepancy?

- We believe that the reviewer's impression of degradation above 12 km stems from Fig. 3, where low-IWC are missing at altitudes above 12 km. Our wording in the discussion may have been misleading; this behaviour reflects poorer performance for low-IWC cases rather than at the specific altitude. These cases are difficult to detect due to ICI's low sensitivity to thin cloud, and this result is independent of altitude, as seen in the lower left corners of all panels of Fig. 2.

  In contrast, our conclusion of reliable performance up to 14 km was based on analyses over the entire IWC range, with particular focus on the mid-IWC regime where ICI is most sensitive. The averaging kernel analysis was performed on a subset of data with IWP $> 0.1$ kg m$^{-2}$ and IWC $> 0.01$ g m$^{-3}$, thus excluding thin cloud cases. The kernels therefore remain stable not despite, but partly because of, the exclusion of low-IWC cases.

- **Changes to the manuscript:** The discussion around Fig. 3 has been updated to better clarify the potential reason for missing clouds in Fig. 3.

In the averaging kernel experiment, Dme response function is bi-model, but IWC response function is not. Do you know why they are inconsistent? Does this suggest ICI is mostly useful for sensing ice particles in anvils and cloud bottom? But the vertical resolution of 5 km at 10 km strikes me... Please elaborate your thoughts.

- Not too much importance should be placed on the exact value of the measurement response. Values near 1 indicate retrievals largely based on data, while values near 0 indicate strong a priori influence. For this reason, we would hesitate to conclude that the measurement response implies usefulness for only anvils and cloud bottoms. Even at mid-altitudes (e.g. 7.5 km), the response remains high (0.85-0.9), showing that the retrieval is still useful.

  Although we cannot fully explain the measurement response shape, similar oscillations appear in classical OEM retrievals (Forkman et al., 2016), typically when the effective vertical resolution is high relative to the altitude spanned by the measurement response. However, we realise that referring to the measurement response as a 'response function' may lead to over-interpretation. Rather, it is defined per level, and should not be labelled bimodal as if it were a continuous function.

  Regarding the resolution of 5 km at 10 km, the poorer resolution compared to IWC is an expected result, since more information is needed to constrain particle size than the location of ice. Also, the resolution estimate summarises *average* behaviour across a range of cloud types, as the reviewer rightly noted in an earlier comment, and does improve for only lower-IWP cases.

- **Changes to the manuscript:** References to the 'response function' have been updated to the 'measurement response'.

Line 170: why using the mean instead of the peak of the predicted PDF? The PDF could be very skewed for many cases.

- We agree that the predicted PDFs are often skewed, which motivates our use of a method that retrieves non-Gaussian PDFs. We chose to use the mean as our final retrieved value since it is a measure of the full PDF and thus captures any skewness. As a result of the reviewer's earlier point on quality flagging, there exist cases where the 99th-1st quantile spread is large, indicating a strong skew and therefore a difference between the mean, median, and peak. We have also checked retrievals using the median of the distribution and found systematic underestimation of IWC. Skew in the higher-IWC end of the PDF therefore pulls the mean correctly upward, showing the importance of capturing the full PDF.

  That said, we recognise the merit of the reviewer's point. In our ongoing work on retrievals based on real satellite observations, we are evaluating the use of the PDF peak. However, small numerical issues arise when constructing the PDF from the retrieved CDF. This instability necessitates smoothing the PDF, which complicates taking the peak as the estimate.

We appreciate the reviewer's interest in this choice, which motivates our ongoing exploration of the two approaches.

5b vs. 5a: I can understand why your training database has low bias near the tropopause because your database doesn't include the CALIPSO measurements while DARDAR has. But why your PBL cloud ice are also low-biased compared to DARDAR?

- We assume the reviewer refers to the discrepancy at 0-1 km in Fig. 5a and 5b, where our mean IWC appears slightly high-biased (perhaps a mistype in the reviewer's comment). This bias arises from the way we treat the radar clutter zone during the radar reflectivity inversions. Specifically, directly above the surface in the radar clutter zone, the reflectivity values are rejected, and the value directly above the clutter zone is copied to the altitudes below. This leads to slightly higher near-surface cloud ice compared to DARDAR.

8a: There is a consistent and significant low-bias in the 1:1 correlation, but then the MFE is very small, indicating the retrieval results are good. Why? Is MFE a good measure for such a situation? Maybe you should use RMSE or MAE?

- We thank the reviewer for this helpful comment, which flagged potentially misleading aspects of our error summaries. We have revisited our use of MFE for $D_{\mathrm{m,IWC}}$ and agree that it may misrepresent retrieval accuracy at higher values. Following the reviewer's suggestion, we calculated the RMSE (expressed as a percentage of the truth). The RMSE shows an increase at higher $D_{\mathrm{m,IWC}}$, which is more inline with expectations. Relative errors remain largest at low $D_{\mathrm{m,IWC}}$, since low values inflate relative RMSE. The trough near $\sim$50 $\mu$m seen in MFE disappears with RMSE, which is also a potentially misleading feature, as it only reflects the median crossing the 1:1 line in a region of low sensitivity. We therefore agree with the reviewer that RMSE is preferable for $D_{\mathrm{m,IWC}}$, but maintain that MFE is a more appropriate metric for IWC due to the several orders of magnitude spanned by this variable.

- **Changes to the manuscript:** Fig. 8 has been updated to use the RMSE instead of MFE. The discussion has been updated accordingly.

**References**

Forkman, P., Christensen, O. M., Eriksson, P., Billade, B., Vassilev, V., and Shulga, V. M.: A compact receiver system for simultaneous measurements of mesospheric CO and O$_3$, Geoscientific Instrumentation, Methods and Data Systems, 5, 27–44, https://doi.org/10.5194/gi-5-27-2016, 2016.

---

## Author Response (AR2)

**Response to Referee #2**

**November 20, 2025**

We thank the reviewer for their additional comments. We have both provided a further analysis in our response here and updated the text in response to their concerns.

L160: "A new noise was generated" -> "A new noise value was randomly generated...". Just to clarify: did you use a Gaussian noise generator with standard deviation of NeDT?

• Yes, this is correct. The text has been updated to include the sentence: 'Each noise value was generated using a random Gaussian noise generator with standard deviation of  $0.75 \text{NE}\Delta\text{T}$ .'

L378: higher -> larger or coarser.

• Fixed.

Fig. 6d: is alarming. And I somewhat disagree with your explanation in Line 506-512. The consistent negative bias for integrated Dm is tied to those small IWC (the artifact in Fig. 6b for those samples with IWP  $< 1.\text{E-}2 \text{ kg/m}^2$ ) I believe.

• We are not certain if the reviewer meant to refer to lines 506-512, since these do not describe Fig. 6d. We therefore focus our response on the bias in the integrated retrieved  $D_{\rm m}$ , rather than the specific lines referred to.

We agree that the negative bias in Fig. 6d warrants an expanded discussion. We have expanded the text in the manuscript, but also respond in more detail here. The bias appears to arise due to several factors, the most influential being the overall tendency to underestimate  $D_{\rm m}^{IWC}$  across its full range, as seen in the retrieval performance figures in Fig. 8. While the mean and median in. Fig. 8 show only a small negative bias, the spread extends considerably lower, particularly at low altitudes and for low  $D_{\rm m}^{IWC}$ . As a result, these underestimations can substantially lower the integrated Dm values and contribute to the negative bias in Fig. 6d.

To illustrate this, we include an example for the reviewer's reference (Fig. 1). It is clear that the region of  $D_{\rm m}^{IWC}$  that contributes most to the integrated  $D_{\rm m}$  is highly underestimated in the retrieval. Interestingly, the low-magnitude artifacts mentioned by the reviewer are also visible in panel (d), but might actually act to reduce a negative bias.

We also find that such cases typically occur for low altitude clouds, which are more prone to underestimation at low  $D_{\rm m}^{IWC}$ . We isolated numerous cases of strong negative bias in the Dm integral, and all showed the same pattern — a localised underestimation of  $D_{\rm m}^{IWC}$  in the region contributing most to the integral. The cases of  $D_{\rm m}^{IWC}$  that are underestimated are not particularly high, and therefore contribute most strongly to the lower end of the distribution in Fig. 6d. These cases also tend to appear when IWP and IWC are relatively low.

For completeness, we also include the distribution of Dm column when all cases are included (Fig. 2), rather than filtering for IWP  $> 10^{-2}$  kg m-2, as done in Fig. 6d in the manuscript. In this case, the truth (integrated database) and retrieval (integrated retrieval) agree much more closely. This indicates that our model reproduces the overall training statistics well, and the filtering of the data unfortunately leads to a negative shift in the integrated Dm distribution. The difference between the integrated database and the database is due to the difference in

vertical resolution between the columns used to calculate each of these variables, as detailed in Sect. 4.3 of the manuscript.

Finally, there is also a negative bias present at the upper end of the  $D_{\rm m}$  range in Fig. 6d. We again attribute this to the underestimation of  $D_{\rm m}^{IWC} > 600~\mu{\rm m}$ , as shown in Fig. 8, which is consistent with the point at which the distributions begin to show stronger disagreement. This is a result also seen in May et al. (2024), when retrieving the column Dm directly.

**• The paragraph referring to Fig. 6d has now been updated in the manuscript, and is as follows:**

'As done for IWP and  $Z_{\rm m}$  in Sect. 5.2, we also compare distributions of  $D_{\rm m}$ , which can be calculated from IWC and  $D_{\rm m}^{\rm IWC}$  according to Eriksson et al. (2020). Distributions are presented in panel (d) of Fig. 6. The distribution of direct retrievals of  $D_{\rm m}$  shows good agreement to the database-derived distribution. For  $D_{\rm m}$  calculated using retrieved  $D_{\rm m}^{\rm IWC}$ , the distribution agrees in shape with the database-derived distribution, but shows larger discrepancies than for  $Z_{\rm m}$ . Agreement is relatively good in the mid-range of  $D_{\rm m}$ . For  $D_{\rm m} < 100~\mu{\rm m}$  and  $D_{\rm m} > 700~\mu{\rm m}$ , poorer agreement is seen. This negative bias is attributed to a general underestino of  $D_{\rm m}^{\rm IWC}$  across most of its range. For example, cases of the integrated retrieved  $D_{\rm m} \sim 50~\mu{\rm m}$  were found to arise due to a strong underestimation of  $D_{\rm m}^{\rm IWC}$  at several altitudes in a profile, typically at low altitudes. These would be cases of relatively low  $D_{\rm m}^{\rm IWC}$  that lie close to or below the 16th quantile in Fig. 8, thus substantially lowering the column integral. Since  $D_{\rm m}^{\rm IWC} > 700~\mu{\rm m}$  is nearly always underestimated, the negative bias at high  $D_{\rm m}$  is likewise expected. Furthermore, unlike  $Z_{\rm m}$  which depends only on retrieved IWC,  $D_{\rm m}$  is derived from both retrieved IWC and retrieved  $D_{\rm m} \sim 50~\mu{\rm m}$  were also typically associated with low IWC, which are retrieved less accurately. Another plausible reason for the differences is that the IWC and  $D_{\rm m}^{\rm IWC}$  errors are correlated at each altitude. However, this information is not provided by QRNN.'

Figure 1: A scene of IWC,  $D_{\rm m}^{IWC}$ , and predicted  $D_{\rm m}^{IWC}$  is shown in panels a, b, and c, respectively. A specific case of integrated  $D_{\rm m}$  that has high negative bias is indicated in panels a, b and c by the dotted white line. The true and retrieved  $D_{\rm m}^{IWC}$  for this specific case is plotted in panel d, and the resulting integrated  $D_{\rm m}$  is given in the legend.

Figure 2: Distributions of  $D_{\rm m}$  calculated from database IWC and  $D_{\rm m}^{IWC}$ , calculated from retrieved IWC and  $D_{\rm m}^{IWC}$ , and a direct retrieval of Dm . The legend in panel (d) applies to both panel (c) and (d). Distributions of Zm and Dm are calculated only for cases corresponding to IWP  $> 10^{-2}$  kg m-2

L615: remove redundant "Fig.".

Fixed.

L798: The error covariance matrices also suggest the effective vertical resolution is  $\sim 2$  to 2.5 km, which is consistent with AK method derived value.

• The following sentence has been added to the 'Summary and conclusions' section: 'The retrieval error correlation matrices also suggest an effective vertical resolution of around 2 to 2.5 km, which is consistent with our estimations derived from averaging kernels.'

**References**

Eriksson, P., Rydberg, B., Mattioli, V., Thoss, A., Accadia, C., Klein, U., and Buehler, S. A.: Towards an operational Ice Cloud Imager (ICI) retrieval product, Atmos. Meas. Tech., 13, 53–71, https://doi.org/10.5194/amt-13-53-2020, 2020.

May, E., Rydberg, B., Kaur, I., Mattioli, V., Hallborn, H., and Eriksson, P.: The Ice Cloud Imager: retrieval of frozen water column properties, Atmos. Meas. Tech., 17, 5957–5987, https://doi.org/10.5194/amt-17-5957-2024, 2024.